# Thermal-optical analysis of quartz fiber filters loaded with snow samples - determination of iron based on interferences caused by mineral dust

Daniela Kau[1], Marion Greilinger[2], Bernadette Kirchsteiger[1], Aron Göndör[1], Christopher Herzig[1], Andreas Limbeck[1], Elisabeth Eitenberger[1], and Anne Kasper-Giebl[1]

[1]Institute of Technologies and Analytics, TU Wien, Vienna, 1060, Austria
[2]Zentralanstalt für Meteorologie und Geodynamik (ZAMG), Vienna, 1190, Austria

*Correspondence to*: Daniela Kau (daniela.kau@tuwien.ac.at)

**Abstract.** The determination of mineral dust and elemental carbon in snow samples is of great interest, as both compounds are known as light absorbing snow impurities. Different analytical methods have to be used to quantify both compounds. Still, the occurrence of mineral dust, which contains hematite, leads to a bias in the quantification of elemental carbon and organic carbon via thermal-optical analysis. Here we present an approach which utilizes this interference to determine the concentration of iron via thermal-optical analysis using a Lab OC/EC Aerosol Analyzer (Sunset Laboratory Inc.) and the EUSAAR2 protocol. For this, the temperature dependency of the transmittance signal determined during the calibration phase, i.e. when all carbonaceous compounds are already removed, is evaluated. Converting the transmittance signal into an attenuation, a linear relationship between this attenuation and the iron loading is obtained for loadings ranging from 10 to 100 µgFe cm⁻². Furthermore, the evaluation of the transmittance signal during the calibration phase allows to identify samples which need to be re-evaluated, as the analysis of elemental carbon and organic carbon is biased by constituents of mineral dust. The method, initially designed for snow samples, can also be used to evaluate particulate matter samples collected within the same high alpine environment. When applying the method to a new set of samples it is crucial to check whether the composition of iron compounds and the sample matrix remain comparable. If other sources than mineral dust determine the iron concentration in particulate matter, these samples cannot be evaluated with thermal-optical analysis. This is shown exemplarily with data of particulate matter samples collected in a railway tunnel.

## 1 Introduction

Black carbon (BC) and mineral dust (MD) have been identified and evaluated as prominent light absorbing snow impurities (LASI) (Doherty et al., 2016; Kaspari et al., 2014; Svensson et al., 2018; Tuzet et al., 2017; Warren and Wiscombe, 1980). Deposition of these compounds leads to a reduction of the surface albedo, influencing thawing and geochemical processes, accelerating snow and ice melt and triggering albedo feedback (Ramanathan and Carmichael, 2008, Tuzet et al., 2020).

The presence of MD in snow can be determined via gravimetric measurements (e.g. Kuchiki et al., 2015), particle counting methods (e.g. Di Mauro et al., 2015) or is calculated based on the concentrations of iron, calcium, aluminum, silicon and titanium (Li et al., 2017).

Quantification of BC concentrations in snow and ice has been done via several methods, which are addressing either BC or elemental carbon (EC). Thermal-optical analysis (Birch and Cary, 1996; Chow et al., 1993), optical methods determining the spectral absorption of total particulate matter sampled on a filter (Grenfell et al., 2011) and laser-induced incandescence (Schwarz et al., 2012) are commonly applied. Regarding the differentiation between BC and EC we refer to the thorough discussion in Petzold et al. (2013). Reviews and comparisons of advantages and disadvantages of the respective methods can be found in literature (Kang et al., 2020; Lim et al., 2014; Qian et al., 2015). Within this work we will focus on thermal-optical analysis (TOA), which allows to differentiate between organic carbon (OC) and EC and represents the reference method for analysis of OC and EC in particulate matter samples (DIN EN 16909:2017). The filter is heated in an inert (helium) and

subsequently in an oxidizing (helium/oxygen) atmosphere, while the evolving carbon is converted to methane and determined with a flame ionization detector. To account for pyrolysis and to separate OC and EC precisely, the transmittance or reflectance of the filter is monitored. The time when the transmittance (or reflectance) reaches its initial value determines the split point to differentiate between OC and EC. Several temperature protocols were introduced to optimize the method (e.g. Cavalli et al., 2010; Chow et al., 2007). Liquid samples can be analyzed after filtration, when water insoluble OC (WinsOC) and EC are

retained on a quartz fiber filter. Previous works address uncertainties related to the filtration process like an undercatch during the filtration, uneven filter loadings or the loss of EC within vessels used for melting and filtration (Forsström et al., 2013; Kuchiki et al., 2015; Lim et al., 2014; Meinander et al., 2020; Torres et al., 2014; Wang et al., 2020).

Uncertainties related to TOA itself apply to both, the analyses of filter residues of liquid samples and the analyses of particulate matter samples, even though there might be some basic differences between the two types of samples, such as a differing

particle size distribution or a slightly modified chemical composition. As we focus on the analytical procedure of TOA, we will avoid the term WinsOC throughout the manuscript and stick to OC. The concept of analyses and evaluation can be applied to the analysis of both sample types, i.e. particulate matter filters (then OC is addressed) or filters loaded with liquid snow samples (then WinsOC is addressed).

The OC/EC split and thus the quantification of both compounds, EC and OC, is influenced by the overall chemical

composition. When elevated loadings of MD are present the influence of carbonates (Karanasiou et al., 2011; Li et al., 2017) and metal oxides (Bladt et al., 2012; Bladt et al., 2014) is important, as MD is rich in those compounds (e.g. Kandler et al., 2007). Here we focus on iron oxides, which are the main light absorbing minerals in MD and cause its reddish (hematite) or yellowish (goethite) color (Alfaro et al., 2004). Formenti et al. (2014) found Fe oxides (goethite and hematite) to account for 2-5% of MD by mass, investigating the mineralogical composition of mineral dust in western Africa. At elevated temperatures,

between 250 and 600°C, goethite changes to hematite (Liu et al., 2013). This temperature is exceeded already during the first part of TOA (helium atmosphere), making hematite the main Fe oxide present on the filters during the time when the split point for OC/EC differentiation is set. Thus, iron oxides do not only determine the properties of MD as a prominent component of LASI, but they also affect TOA. To exceed the method's limit of detection for OC and EC, filtration of a rather large amount of liquid snow or ice is desirable. By doing so, MD, containing iron oxides, is enriched and interferences occur. Wang et al.

(2012) used TOA for analyses and identified an extra decrease in optical reflectance during the 250°C heating stage in hematite and suggested to shift the reference value for the OC/EC split to this temperature step. Gul et al. (2018) suggested to enhance these adjustments even further, defining the reference value for the OC/EC split at 550°C for filters with higher MD loads.

In this work we investigate the influence of MD loads on TOA of snow samples collected in a high alpine environment. The main interest of our work is to investigate the temperature dependence of the light attenuation caused by Fe containing

compounds in MD. This led us to a new approach to approximate Fe concentrations via TOA. Therefore, we evaluate the transmittance signal during the calibration phase, i.e. when carbonaceous compounds are no longer present. The method is tested extensively for snow samples and evaluated briefly for particulate matter samples.

## 2 Collection of samples and sample preparation

### 2.1 Snow samples

Surface snow samples were collected on the upper platform of the global GAW (Global Atmosphere Watch) station Sonnblick Observatory, situated at 3106 m a.s.l., in the Austrian Alps. MD deposited on Mount Sonnblick often derives from long range transport of dust from desert regions (Greilinger et al., 2018). For sampling, the uppermost centimeters (approximately 5 cm) of snow were collected directly in Whirl-Pak™ bags (Nasco™). For this purpose, the bag was opened and the opening, which is reinforced by a wire handle, was used as a shovel to scoop in the top snow layer. Samples with and without a visible influence

of MD were collected during multiple visits at the station between 2018 to 2020 and were kept frozen until further processing.

In the laboratory, samples (approximately 600 g each) were melted gently in a glass beaker using a microwave (600 W). Aliquots of the liquid samples were filtrated onto precleaned (24 h at 600°C, cooling in a desiccator above distilled water) quartz fiber filters (Pallflex® Tissuquartz™; loaded area was circular with a diameter 16 mm, half used for analysis) applying a slight vacuum. After filtration, these snow sample filters were dried overnight in a desiccator above silica gel to allow further analysis.

Blank filters were prepared in a similar way, filtrating ultrapure water instead of the liquid snow sample.

## 2.2 PM$_{10}$ samples

PM$_{10}$, i.e. particles with an aerodynamic diameter equal to or less than 10 µm, was sampled onto quartz fiber filters (Pallflex® Tissuquartz™) in the same high alpine environment as the snow samples using a high volume sampler (DIGITEL Elektronik AG, Switzerland; sampling duration: 1 week) and in tunnels with railway traffic using a low volume sampler SEQ47/50 (Sven Leckel Ingenieurbüro GmbH, Germany; sampling duration: 4 h). No precleaning step of the filters was performed before sampling. Rectangular aliquots with an area of 1.5 cm² were used for analysis.

## 2.3 Reference samples

Hematite (Fe$_2$O$_3$) was chosen as a reference substance for light absorbing compounds in MD as goethite and hematite are the most abundant forms of Fe oxides in MD (Formenti et al., 2014) and goethite will be converted to hematite at elevated temperatures starting at 250°C and being completed at 600°C (Liu et al., 2013), i.e. at temperatures which correspond to the conditions used in the inert phase of TOA already. Fe$_2$O$_3$ was suspended in ultrapure water (8.75 mg in 1 L flask) in an ultrasonic bath. Quartz fiber filters were loaded with different amounts of the suspension to cover the range of the Fe loadings found on filters loaded with snow samples containing MD (roughly 9 to 314 µgFe cm$^{-2}$). The procedure of suspending the particles and loading the filters was done on two different days to verify reproducibility.

A different set of samples was prepared with Standard Reference Material® 2709 (SRM 2709 San Joaquin Soil, National Institute of Standards and Technology, USA). SRM 2709 was suspended in ultrapure water (83.3 mg in 1 L flask) and quartz fiber filters were loaded covering the range of roughly 3 to 141 µgFe cm$^{-2}$.

## 3 Instrumentation

### 3.1 Thermal-optical analysis

Thermal-optical analysis (TOA) is the reference method for the determination of OC and EC in ambient aerosols (DIN EN 16909:2017). Filter aliquots are heated in an inert and oxidizing atmosphere, the evolving carbonaceous compounds are converted to methane which is quantified via a flame ionization detector. To correct for pyrolytic carbon, the transmittance of the filter aliquot is recorded. As explained in detail in the results section, the development of the transmittance signal is influenced by the Fe loading of the filter. Therefore, we solely investigate this transmittance signal within this manuscript. A Lab OC-EC Aerosol Analyzer (Sunset Laboratory Inc., USA) and the EUSAAR2 protocol (Cavalli et al., 2010; temperature steps are given in Table S1 in the Supplement) were used. The instrument logs the transmittance and reflectance of the sample at a wavelength of 660 nm during the measurement. For the measurement, the program OCEC834 and for the evaluation of the raw data the program Calc415 (both Sunset Laboratory Inc., USA) were used. Further processing of the transmittance data was conducted with an external program (Microsoft Excel, Microsoft Corporation, USA).

### 3.2 Inductively coupled plasma-optical emission spectroscopy and -mass spectrometry

Inductively coupled plasma-optical emission spectroscopy (ICP-OES) and inductively coupled plasma-mass spectrometry (ICP-MS) were used to quantify Fe. For this, a microwave assisted digestion of the filter aliquots was conducted subsequent

to TOA. Filters loaded with snow samples or reference materials ($Fe_2O_3$, SRM 2709) samples and high alpine $PM_{10}$ samples were digested using a microwave system (Multiwave 5000, Anton Paar, Austria) and METHOD 3052 (method for microwave assisted acid digestion suitable for siliceous, organic and other complex matrices; US EPA, 1996; $HNO_3$:HCl:HF 6:2:1, maximum temperature: 180°C) and were analyzed for Fe via ICP-OES (iCAP 6500 ICP-OES spectrometer, Thermo Scientific, USA). Filter aliquots of the particulate matter samples collected within the railway tunnel were digested using a microwave system (Multiwave 3000, Anton Paar, Austria/Start1500, MLS GmbH, Germany; aqua regia, maximum temperature: 220°C) and Fe was quantified via ICP-MS (iCap Q System instrument, Thermo Scientific, USA). The limit of detection was 0.4 µg cm⁻² for ICP-OES and 0.1 µg cm⁻² for ICP-MS.

### 3.3 X-ray powder diffraction

X-ray powder diffraction (PXRD) was used for specification of Fe present on the snow sample filters. Experiments were carried out in Bragg Brentano geometry using an Empyrean diffractometer (Malvern PANalaytical B.V., Netherlands; scattering angle range of $5° < 2\theta < 135°$). A focussing mirror was used to provide Cu $K_{\alpha1,2}$- radiation for the experiment. The beam divergency was defined by using a 1/4° fixed vertical entrance slit followed up by a 0.04 rad horizontal Soller slit and a 0.04 rad horizontal Soller slit on the secondary side in front of an open line detector (GaliPix detector). The detector to sample distance for this instrument was fixed to 240 mm.

The PXRD diagrams were evaluated using the Malvern PANalytical program suite HighScorePlus v4.6a (Degen et al., 2014). A background correction and a $K_{\alpha2}$ strip were performed. Crystallographic phases were assigned based on the ICDD-PDF4+ database (Kabekkodu et al., 2002).

### 3.4 Scanning electron microscopy

Scanning electron microscopy (SEM) measurements allowed to visualize particle sizes and their elemental composition. Thus, a comparison of filters loaded with snow samples and the reference substances $Fe_2O_3$ and SRM 2709 was possible. A FEI Quanta 200 (Thermo Fisher Scientific, USA) instrument equipped with an Octane Pro EDS System (EDAX, USA) was used for the analyses of loaded quartz fiber filters. Samples were coated with Au prior to analysis.

### 4 Results

### 4.1 Transmittance of filters loaded with snow samples during TOA

During TOA, the transmittance of the filters changes due to pyrolysis and removal processes of carbonaceous compounds, resulting in a clean filter after the analysis is completed. However, if MD containing colored Fe compounds, e.g. hematite, is present, the filters remain colored and the transmittance stays lower than for a clean filter, even after the complete removal of carbon. Thus, the transmittance, and respective changes of this value observed throughout TOA, can either be due to the carbon load or constituents of MD. The influence of the various compounds becomes visible when the same filter is analyzed repeatedly, as illustrated in Fig. 1. The change in transmittance of a filter loaded with a snow sample without visible MD contamination during TOA is given in Fig. 1a, while Fig. 1b shows the results obtained for a snow sample filter containing MD. For both samples, multiple measurements (three to five reruns) of the same filter were conducted to separate the change in transmittance deriving from carbonaceous compounds and from MD, which is left on the filter as a residue after the first analytical run. The course of the transmittance signal within the first run, when carbonaceous compounds are still present, is given as a black line. The average course of transmittance of the reruns is shown as a grey line, whereas the standard deviation of the reruns is plotted as a grey area. The reported temperature is the temperature inside the sample oven. The vertical lines denote the switch from an inert (He) to an oxidizing atmosphere (He/$O_2$) and the start of the calibration phase (Cal). Before introducing oxygen and starting the second temperature profile, the temperature is reduced temporarily. At the end, the

thermogram includes a cooling phase. At that time, calibration is performed. The part of this cooling phase, which will be used for further evaluations, is highlighted in orange. The Fe loading on the filter loaded with MD was 30 µgFe cm$^{-2}$, while it was below 2.5 µgFe cm$^{-2}$ for the sample without visible MD contamination. This corresponds to Fe concentrations in the liquid snow sample of 1.1 mgFe L$^{-1}$ (sample containing MD, 54 mL of snow filtrated) and 51 µgFe L$^{-1}$ (sample without MD, 99 mL of snow filtrated). The total carbon loads on the filters were 66 µgC cm$^{-2}$ (sample containing MD) and 28.5 µgC cm$^{-2}$ (sample without MD), but for better clarity, the FID signal is not shown in any panel of Fig. 1.

Both samples, with and without MD, show pronounced changes of the transmittance signal during the first analysis represented by the black line. Changes are more pronounced in Fig. 1b, which can be expected due to the higher total carbon load. For the sample without MD (Fig. 1a), the split point could be set automatically and was reached during the He/O$_2$ 3 phase, i.e. the third temperature step during the oxidizing phase. For the sample containing MD (Fig. 1b), the initial transmittance was never reached, and thus no EC would be determined, while OC would be overestimated. Applying the approach described by Wang et al. (2012), i.e. using the transmittance at 250°C as a reference if MD is present, still no split point can be determined before the cooling phase starts. Using the approach of Gul et al. (2018) and taking the transmittance at 550°C as a reference, an identification of EC becomes possible. For completeness, we added a thermogram showing the FID signal of the sample containing MD in the Supplement (Fig. S1).

During the reruns (grey line) only minor changes in the transmittance signal are determined for the sample without MD (Fig. 1a), which is in good agreement with the recordings obtained for the blank filter (Fig 1c). The sample containing MD (Fig. 1b) behaves differently. Pronounced changes of the transmittance are visible for the reruns as well. The transmittance decreases whenever the temperature increases, and vice versa. Obviously, these changes of transmittance are due to constituents of the remaining MD and not induced by pyrolysis or combustion of carbonaceous compounds. As hematite has already been identified as the main light absorbing compound in MD, several filters were loaded with hematite and analyzed as reference samples. To confirm the suitability of hematite as a reference compound, PXRD measurements were performed before and after TOA for a subset of the filters loaded with snow containing MD. Hematite was clearly identifiable in all samples, but due to the high background of the filter matrix and the rather low dust loadings a more detailed specification of hematite could not be obtained. The changes in transmittance of Fe$_2$O$_3$ during TOA are shown exemplary for a filter loaded with 19 µgFe cm$^{-2}$ in Fig. 1d. Again, replicate analyses (three reruns) were performed. Results of the first run are not shown, as marked differences of the transmittance signal were observed between the first measurement and the reruns, which could be attributed to the evolution of carbonaceous material attached to the hematite (total carbon approx. 5 µgC cm$^{-2}$). The procedure to omit the first run is in accordance with Wang et al. (2012) and Yamanoi et al. (2009), who also preheated the hematite once before further analyses were conducted. The change in the transmittance signal of hematite compares well to the results of the snow sample containing MD (Fig. 1b). Again, an opposing trend of transmittance and temperature was observed. The variation spans an overall range of approximately 500 a.u. and the minimum values are found at highest temperatures. The variation of transmittance which can be attributed exclusively to refractory and light absorbing compounds in MD is lower than the variation observed during the first run of analyses, when carbonaceous compounds are present. Still, it is large enough to markedly affect the determination of the split point and explains why the initial value of transmittance cannot be reached at all, when MD loadings are high.

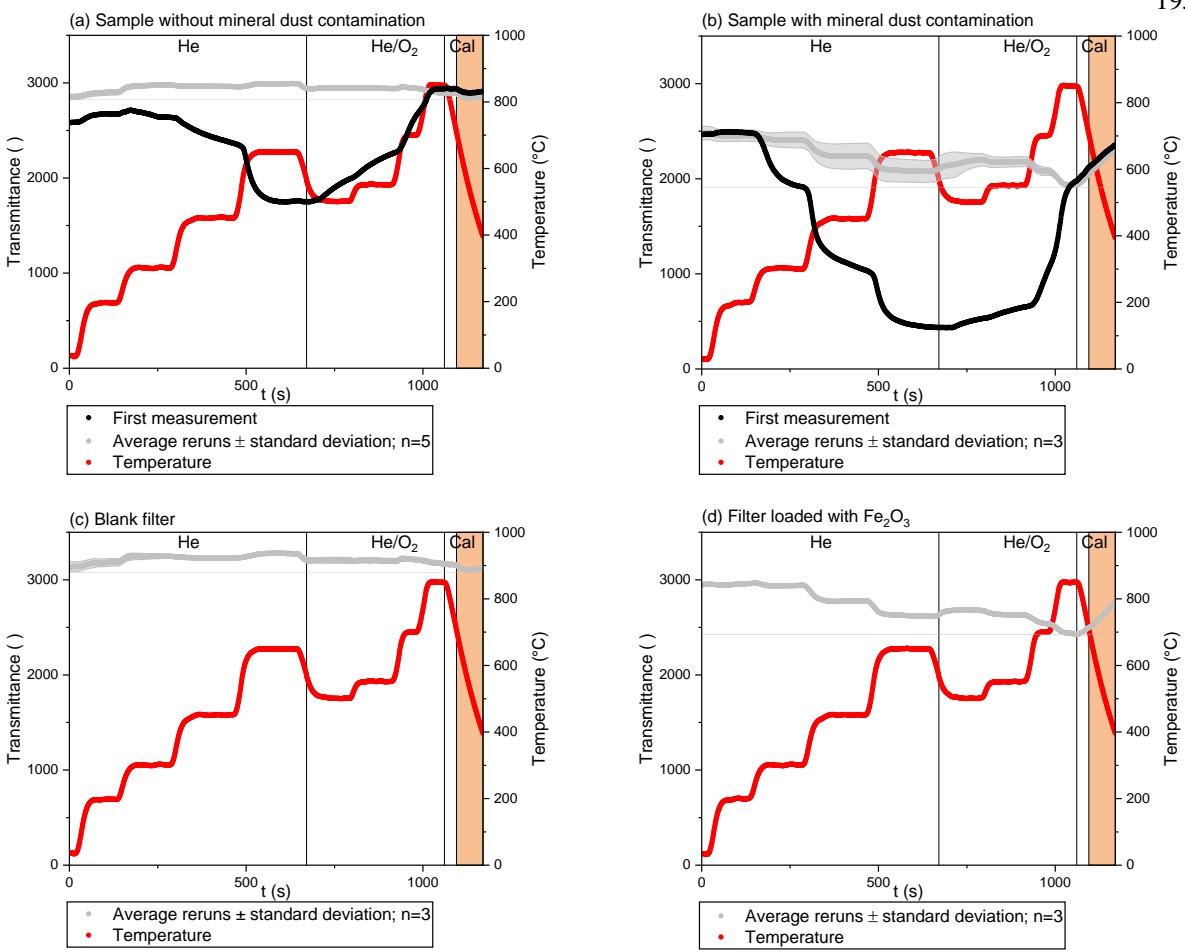

**Figure 1: Transmittance during TOA of quartz fiber filters loaded with a snow sample without (a) and with (b) mineral dust contamination, a blank filter (c) and a filter loaded with Fe₂O₃ (d). Details about the filter loadings are given in the text.**

During TOA, transmittance is determined at a wavelength of 660 nm where Fe oxide minerals have a rather low absorbing potential compared to shorter wavelength (Alfaro et al., 2004). Therefore, such a severe interference is not expected. However, these evaluations refer to ambient temperature, while high-temperature visible spectroscopy showed that reflectance spectra and colors of hematite changed markedly with temperatures up to 800°C (Yamanoi and Nakashima, 2005; Yamanoi et al.,

2009). The changes are especially pronounced at longer wavelength (> 550 nm), meaning that the red color of hematite becomes black with temperature increase (Yamanoi et al., 2009). This phenomenon introduces a severe bias to the determination of the split point, as its position can be expected at a temperature above 500°C. At that point the darkening of the filter due to MD will be much more pronounced than at the initial temperature or even at the temperature ranges suggested earlier to correct for a potential bias of the split point (Gul et al., 2018; Wang et al., 2012) and a systematic error in the

determination of EC and OC occurs. Using the transmittance at 250°C for the determination of the split point, the influence of MD still remains underestimated: At the time of the split point the temperature will definitely be higher and thus the reduction of the transmittance signal due to hematite more pronounced. It would be mere coincidence if this effect was offset by an early pyrolysis of OC. Using the transmittance at 550°C accounts better for this phenomenon, however, the difference of the black and the grey lines in Fig. 1b shows that marked amounts of pyrolytic carbon can be formed at this point of the analysis already.

If this is the case it will lead to an overestimation of EC accompanied by an underestimation of OC.

The error in the determination of EC and OC cannot be quantified based on a single run of analysis and we suggest a second run to assess the relative influences of both, pyrolytic carbon and MD. The need to perform a second run of analyses can be derived from the readings of the transmittance signal during the calibration phase. At this point of the analysis, all carbon is

already removed, a wide temperature range is covered and the gas flow through the main oven maintains oxidizing conditions, like in the second part of TOA when the split point is set. As expected, the transmittance signal is rather constant for the sample without MD contamination (orange part of the graph in Fig. 1a) and compares well to the transmittance signal of the blank filter (orange part of the graph in Fig. 1c). Samples containing MD show a marked increase in transmittance during the calibration phase, i.e. when the sample is cooling (Fig. 1b). The same effect is visible for the reference sample of hematite in Fig. 1d. The change in transmittance during the calibration phase is always visible, during the initial measurement of the sample and for the reruns. Thus, a decision for an additional run of TOA can be based on the calibration phase of the first run. If a marked increase in transmittance is visible, an interference has to be expected and we recommend to reanalyze the sample to set the split point more precisely. As explained in more detail later, a clear dependence of the transmittance signal on the temperature was observed for samples with Fe loadings above $10\,\mu gFe\,cm^{-2}$. A detailed description of a correction procedure is beyond the scope of this work, which focuses on the temperature dependence of the transmittance signal and its relationship to Fe concentrations.

For completeness we want to mention that high loads of MD might have other influences as well, as precombustion of EC might occur in the presence of inorganic compounds contained in MD, as was shown by Bladt et al., (2012) and Neri et al., (1997).

## 4.2 Using TOA for analysis of Fe in reference samples

The evaluation of the transmittance signal logged during the last 75 s of the calibration phase offers more possibilities than a qualitative judgement whether MD might interfere with the OC/EC split point. The sample temperature in this interval changes between slightly above 700°C to slightly below 400°C. To consider the different loadings of the filters, the transmittance data was converted to an attenuation (ATN), defined in Eq. (1).

$$ATN = 100 * \ln\left(\frac{I_0}{I}\right) \quad (1)$$

This conversion follows the approach by Nicolosi et al. (2018), who defined I as the transmittance signal at a particular time step and $I_0$ as the laser transmittance at the end of the TOA, which reflects an unloaded filter. In our case $I$ refers to a particular temperature rather than to a time step. Furthermore, $I_0$ is more difficult to determine, as the initial transmittance of the unloaded filter is not reached at the end of the measurement, when the filter remains loaded with light absorbing compounds like hematite. Using the transmittance signal at the end of the calibration phase as reference value $I_0'$, the calculated ATN becomes a relative value, expressing the changes relative to the transmittance at 400°C. This value will always be smaller than the actual ATN of the light absorbing material. Calculating $ATN_{700-400}$ based on the transmittance values at 700°C ($I$) and 400°C ($I_0'$) we approximate color changes of Fe containing compounds, e.g. hematite, in this temperature interval. First, we evaluate the transmittance signal of filters loaded with two reference substances, $Fe_2O_3$ and SRM 2709. Plotting the Fe loading of the filter expressed as Fe ($\mu g\,cm^{-2}$) versus $ATN_{700-400}$ (Fig. 2) shows a similar trend for SRM 2709 and pure hematite. $ATN_{700-400}$ values increase up to Fe loadings of approximately $150\,\mu gFe\,cm^{-2}$. At higher loadings $ATN_{700-400}$ starts to decrease again. This effect is due to the definition of $ATN_{700-400}$ as a relative value, i.e. the difference between the actual ATN at 400°C ($ATN_{400}$) and at 700°C ($ATN_{700}$). $ATN_{400}$ and $ATN_{700}$ relate the transmittance at the respective temperatures to the transmittance value of the unloaded quartz fiber filter at room temperature ($I_0$). These values could only be approximated, as we did not determine the transmittance values of the unloaded filters. The respective data is presented in the Supplement (Fig. S2). As could be expected for filter-based measurements, a linear relationship between Fe loadings and ATN could only be determined for low Fe values, but saturation occurs at higher loadings (Gundel et al., 1984). The onset of this effect is different for $ATN_{400}$ and $ATN_{700}$, i.e. it starts at lower Fe loadings for $ATN_{700}$. Calculating $ATN_{700-400}$ as the difference of $ATN_{700}$ and $ATN_{400}$ leads to the trend shown in Fig. 2, and to the unfavorable situation that a defined $ATN_{700-400}$ can be attributed to two Fe loadings. However, this

effect is of minor importance for actual analysis. Fe loadings above 150 µgFe cm$^{-2}$ are avoided, as these filters are prone to sample loss.

We attribute the differences between the filters loaded with SRM 2709 and Fe$_2$O$_3$ shown in Fig. 2 to differences in the overall loadings of the filters (i.e. the loading with other material than hematite) and the particle sizes. Such differences are confirmed via SEM analyses of selected filters loaded with both reference materials. While Fe$_2$O$_3$ filters only contained hematite, SRM filers showed different kinds of minerals, i.e. particles consist of Ca, K, Si, Al, Mg, Na and O and just a minor contribution of Fe, which obviously leads to lower ATN values. Hematite particles on the Fe$_2$O$_3$ filters were generally larger in size compared to Fe-rich particulate matter in SRM 2709. Examples of the SEM analyses, elemental maps and spectra are given in the Supplement (Fig. S3).

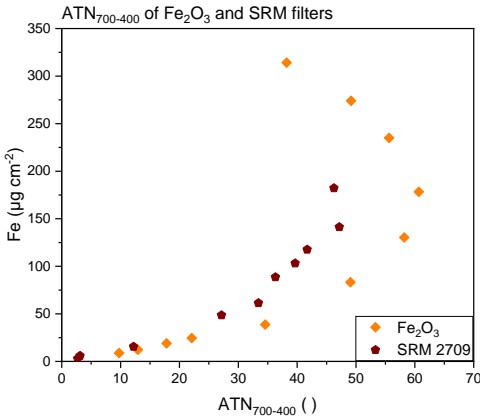

**Figure 2: Fe loading of quartz fiber filters loaded with Fe$_2$O$_3$ or SRM 2709 determined via ICP-OES plotted versus ATN$_{700-400}$.**

### 4.3 Applying TOA for the determination of Fe in snow and PM$_{10}$ samples

Applying the approach described above to filters loaded with particulates in snow, a similar trend is obtained, although the snow samples filters generally give smaller values of ATN$_{700-400}$ (Fig. 3) compared to the measurements with pure hematite or SRM 2709 (Fig. 2). ATN$_{700-400}$ values for snow sample filters are better comparable to the filters loaded with SRM 2709 than those loaded with Fe$_2$O$_3$. The chemical composition of particulates in snow is more similar to SRM 2709 than to pure hematite, as shown in Fig. S3 in the Supplement. Still, SEM images show differences in particle size of SRM and snow filters, which might induce the differences together with differences in the chemical composition, which cannot be identified via SEM.

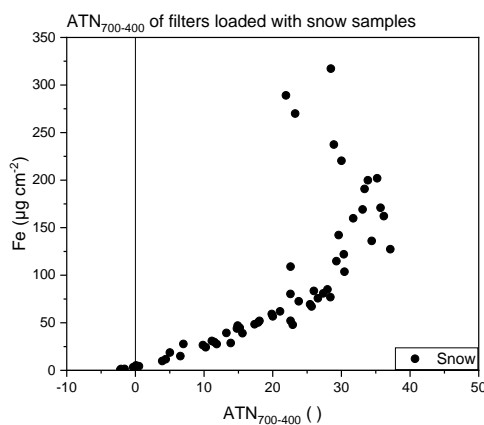

**Figure 3: Fe loading of quartz fiber filters loaded with snow samples determined via ICP-OES plotted vs. ATN$_{700-400}$.**

Restrictions have to be set to define a range which allows a definite allocation of ATN$_{700-400}$ to an Fe loading. It is easily possible to distinguish between samples of the upper and lower part of the curve, as Fe loadings above 150 µg cm$^{-2}$ always show transmittance values below 600 a.u. at a temperature of 400°C in the calibration phase. Thinking of actual Fe loadings

on snow sample filters, Fe loadings above 150 µg cm$^{-2}$ are hardly found, as already mentioned above. Additionally, an upper and lower limit for ATN$_{700-400}$ values was set. The upper limit (i.e. ATN$_{700-400}$ < 29) excludes values with higher variation, while the lower limit (i.e. ATN$_{700-400}$ > 3.5) accounts for samples without MD, which do not show a distinct relationship between the transmittance and the sample temperature (see Fig. 1). Hence, these samples show very low or even negative values of ATN$_{700-400}$. Based on the mentioned restrictions (transmittance > 600 a.u. at 400°C in the calibration phase and 3.5 < ATN$_{700-400}$ < 29) Fe loadings reaching from 10 µg cm$^{-2}$ to 100 µg cm$^{-2}$ become accessible via TOA.

The selected data points are presented in Fig. 4. A fit was computed applying the least square approach to calculate Fe loadings based on the ATN$_{700-400}$ values. The resulting function is given in Eq. (2) and features a coefficient of determination R$^2$ of 0.9691. The residuals did not show distinctive features. The difference between the calculated and the measured Fe loadings divided by the measured Fe loading was below 10% for 20 of 34 samples, above 25% for 7 samples, the maximum being 41% and the median 8%.

$$Fe(\mu g\ cm^{-2}) = 2.9144 * ATN_{700-400} \quad (2)$$

Until now we only addressed filters loaded with insoluble material filtrated from snow samples. Of course, MD is also occurring in ambient aerosols and similar effects can be expected for these filters. To check whether the fit is also applicable for PM$_{10}$ samples, ambient air filters obtained at the Sonnblick Observatory were analyzed and evaluated in the same way. Loadings above 10 µgFe cm$^{-2}$ were hardly achieved during the long-term data set, covering June 2017 to March 2022. Only six samples out of the data set showed Fe loadings in the appropriate range. The PM$_{10}$ samples represent events with long range transport of desert dust to the high alpine site and are in good agreement with the model obtained for the snow sample filters. These data points are also included in Fig. 4. Applying the fit calculated for the snow samples, the relative differences between the Fe loadings determined with TOA and with ICP-OES were between 3% and 49% with a median of 24% for the PM$_{10}$ samples.

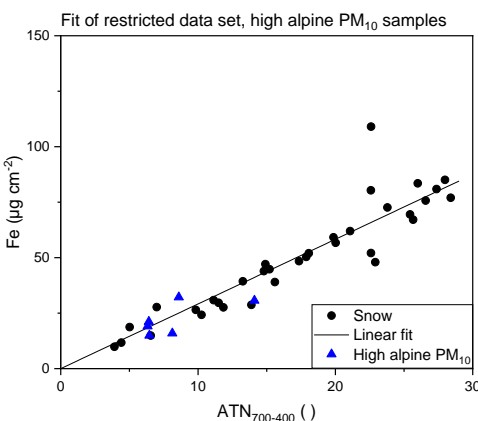

**Figure 4: Linear fit for Fe loading and ATN$_{700-400}$ of quartz fiber filters loaded with snow samples and comparison with high alpine PM$_{10}$ samples showing sufficiently high Fe loadings.**

For snow samples and PM$_{10}$ samples from a high alpine environment, the evaluation of ATN$_{700-400}$ allows the quantification of Fe between 10 and 100 µg cm$^{-2}$ from TOA transmittance data using the same linear approach. Higher Fe loadings can be identified with TOA but should not be quantified. As the snow samples were collected over a time period of two years, a variety of transport events is included in this evaluation. Still, we want to point out that different source regions might lead to differences in the composition and particle size distribution of MD, which might influence such a fit. Nevertheless, the applicability of the fit to at least a limited number of PM$_{10}$ filters sampled in the same high alpine environment is a promising result.

To test whether the approach is generally applicable for aerosol samples with marked contributions of Fe compounds, we included samples collected within a railway tunnel. This represents a completely different environment as very high Fe

contributions and different Fe compounds, like magnetite, hematite and Fe metal, can be expected (Eom et al., 2013), originating from abrasion of tracks, wheels and brakes of the trains. For the tunnel samples included in Fig. 5a, Fe accounts for up to 73% of particulate matter mass, but no specification of Fe compounds has been performed. This high share of Fe in particulate matter mass highlights the different origin of Fe compared to the high alpine samples, as Fe accounts for less than

5% of mass in MD only. 79 filters collected within railway tunnels were evaluated. No correlation between $ATN_{700-400}$ and Fe loadings was found for these samples, although loadings reach up to 325 µgFe cm$^{-2}$. The transmittance at 400°C during the calibration phase was never below 600 a.u., which was the case for the snow sample filters and the filters loaded with hematite at Fe loadings above 150 µg cm$^{-2}$. Only 26 samples show an $ATN_{700-400} > 3.5$ and thus meet the criteria for the evaluation based on $ATN_{700-400}$. All other samples featured values for $ATN_{700-400}$ below 3.5, partly even negative values. Even tunnel

samples with Fe loadings up to 133 µg cm$^{-2}$ showed negative values for $ATN_{700-400}$. All samples are shown in Fig. 5b. "Tunnel $PM_{10}$, Crit" is the subset of samples, which meets the criteria defined for an unambiguous evaluation of Fe containing compounds in snow. While snow samples that do not meet the criteria featured Fe loadings either below 10 or above 100 µg cm$^{-2}$, tunnel samples that do not meet the criteria are found across the whole range of Fe loadings, i.e. 2 to 325 µg cm$^{-2}$. Despite the high Fe loading on the tunnel filters, automatic split points could be set for most of the tunnel samples. The temperature

dependent influence of the Fe loads on the transmittance signal was smaller compared to the changes in transmittance deriving from carbonaceous compounds. This discrepancy between snow samples and tunnel samples is due to the different origin and form of Fe present and could also be noticed via visual inspection of the filters after TOA, as they were differently colored. Photos of two filters are presented in the Supplement (Fig. S4).

The comparison of all sample groups underlines that Fe loadings can only be deduced from the $ATN_{700-400}$ signal when samples

contain Fe in forms as present in MD, e.g. hematite and goethite. If other forms of Fe are expected, the evaluation cannot be applied. Additionally, it shows that samples from the same environment (snow and $PM_{10}$ samples from high alpine environment) show a similar behavior, while MD from different origin (SRM 2709) might need an independent fit to estimate the Fe loading on the filters.

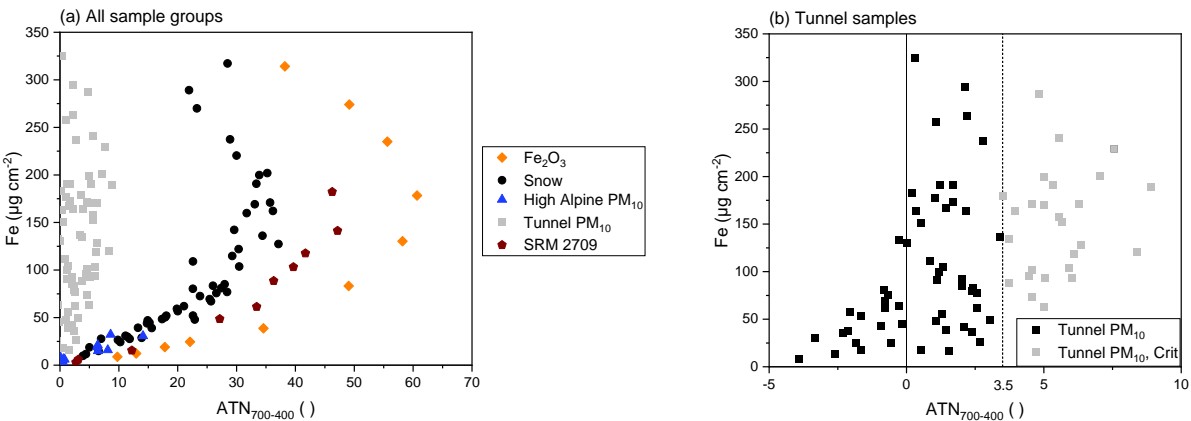

**Figure 5: Fe loadings of all sample groups vs. $ATN_{700-400}$ (a) and Fe loading of $PM_{10}$ samples from tunnel measurements vs. $ATN_{700-400}$ (b).**

All data presented in this work was obtained using the EUSAAR2 protocol, but only the signals determined during the calibration phase are considered. As this temperature range is also covered by the calibration phase of other commonly used thermal protocols, e.g. IMPROVE_A or NIOSH-like protocols, the evaluations remain valid for these protocols as well, as observed for a limited set of samples not discussed here. Note that the evaluation was conducted using the transmittance and not reflectance signal. Post-processing of data already available is possible, as no adjustment of the widely used thermal

protocols is necessary.

## 5 Conclusion

Investigating the influence of MD, i.e. its light absorbing constituent hematite, on TOA, we presented an easily applicable approach to determine Fe loadings on filters containing MD based on temperature changes of the transmittance signal recorded during the calibration phase of TOA. The method is based on the evaluation of quartz fiber filters loaded with snow samples containing different amounts of MD

Considering the whole range of Fe loadings investigated, the relationship between the Fe loading and $ATN_{700-400}$ was shown to be non-linear. By setting two straightforward criteria ($ATN_{700-400}$ between 3.5 and 29 and transmittance signal at 400°C above 600 a.u.) the determination of Fe loadings between 10 to 100 $\mu gFe\ cm^{-2}$ becomes possible based on a linear approach. This approach allows to obtain another compound of LASI with the same method (TOA) as used for the determination of EC. This can be advantageous if the available amount of sample is limited and a proxy for MD is needed. Still a reliable conversion factor of Fe to MD has to be deduced by independent analysis or taken from literature and is not presented here.

The method developed for the analyses of filters loaded with snow samples was successfully applied to the analyses of particulate matter filters originating from the same high alpine environment. We cannot give information about the general applicability to $PM_{10}$ samples loaded with different types of MD, yet. We hypothesize that it should be possible to use the approach as long as MD and its constituent $Fe_2O_3$ remain the dominant source of Fe loadings. Still, an independent calibration might be needed to account for differences in MD composition. Fe loadings in particulate matter samples collected within a railway tunnel and thus having a strong influence of abrasion products from tracks and wheels and not from MD cannot be analyzed with the method presented here. This illustrates the limitation of the method.

Filters loaded heavily with MD (Fe loadings above 10 $\mu gFe\ cm^{-2}$) will experience a severe bias of the OC/EC split point. EC will be underestimated or cannot be determined at all, while OC will be overestimated.

We recommend to rerun the analyses, as soon as the evaluation of the transmittance signal in the calibration phase indicates an influence of MD. This second run of analyses will allow to set the split point more precisely.

### Data availability

$ATN_{700-400}$ values and Fe concentrations of all samples presented in this work are made available in Table S2 in the Supplement.

### Author Contribution

DK and AKG designed the experiments, performed the data evaluation and interpretation and prepared the manuscript. Analyses were conducted by DK (preparation, TOA and digestion of high alpine, $Fe_2O_3$ and SRM samples), BK (evaluation of digestion protocols, quality assurance), AG (TOA and digestion of tunnel samples), CH (ICP) and EE (SEM). MG performed formal analyses and was responsible for the collection of part of the snow samples. Resources were provided by AKG and AL. All authors contributed to reviewing and editing the manuscript.

### Competing interests

The authors declare that they have no conflict of interest.

### Acknowledgements

Special thanks go to Klaudia Hradil for conducting PXRD-analyses at the X-Ray Center at TU Wien, Austria, and for her valuable input. The authors acknowledge TU Wien Bibliothek for financial support through its Open Access Funding Program.

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
