# Peer review of "Thermal-optical analysis of quartz fiber filters loaded with snow samples - determination of iron based on interferences caused by mineral dust"

_Atmospheric Measurement Techniques, 2022_

## Referee Comment (RC1)

Referee report on Kau et al. 'Thermal-optical analysis of snow samples – challenges and perspectives introduced via the occurrence of mineral dust.'

General comments

This manuscript presents a new approach in dealing with the interference of mineral dust (hematite in particular) that might occur during thermal-optical analysis of filter substrates. The paper would be highly welcomed in the literature, due to the reoccurring phenomenon of mineral dust disturbing OCEC-analyzes. Since it is a novel approach, the text is well suited for AMT. In its current form, however, the paper needs some substantial work. Currently, the full potential of the manuscript has not been explored, but rather, corners appear to have been cut, resulting in several sections that either lack or provide minimal information. As an example, the approach is tested for high alpine PM10 filter samples, but only two samples are available. More data would be needed to provide a more robust basis for the authors to discuss the results on that topic. Is there data available data from another site (other than Sonnblick?). Similarly, the snow samples from Sonnblick could be compared with snow samples from another location. This would mean more work, but in the end, it would only and built-up the discussion/conclusions, and ultimately, strengthen the manuscript. With that said, I would reiterate that major revisions are necessary for this paper that could have great potential breakthrough in this area of science. Please see more comments below.

Major comments

Section 3. The different instrumentation used in the study seem applicable and impressive. Still, it is difficult for a reader to see their place and role in the different steps of the approach. It could easily be misinterpreted that the instruments were just randomly chosen when the authors clearly thought about how to best utilize their instruments for this method. One suggestion is to provide an introductory section before each instrumentation is expanded upon. In this introductory text, it should be explained why the instrument is used. In a way, it should end up as a flow-chart explaining the different steps of the analysis, providing future readers with a roadmap of the approach enabling them to easier follow along.

The aims of the paper should be scrutinized. Currently, there is a mismatch between the aims (denoted as interests in the introduction) stated in the introduction and once they are re-stated in the conclusions. The aims should be clearly reviewed and revised in the introduction. On the whole, the conclusions that are discussed particularly in lines 298-308 are not supported by the data currently presented in the paper (e.g. recommending a rerun of analyses; that the approach was successfully applied to PM10 filters also, see additional comment on this below).

Section 4.2 and Fig.2. For the samples contaminated with the reference hematite, was it one batch of filter samples? In other words, was this step ever repeated, with another independent set of hematite contaminated filters? In order to test whether the same pattern would be repeated in the filter samples this is a suggestion of work to be done. Similarly, what about conducting experiments with reference filters

containing Fe originating from a different source than hematite? This could further confirm the claims made that this method works well for hematite by not other types (which currently is not strongly supported by much data; only the tunnel samples).

Section 4.3. It is understood that there are only two $PM_{10}$ samples from Sonnblick. Yet, two samples are not enough to support the claims the authors want to make in the text (concerning the applicability of the fit for $PM_{10}$ samples). The authors need to find a way to expand with more samples for the $PM_{10}$ or majorly down-play the importance of only two samples in the claims they are making.

Minor comments

Title. As it currently reads the title of the paper suggests that *challenges* and *perspectives* are to be discussed. The use of such wording would be more appropriate if this was a review article. I would encourage the authors to critically think through the title and modify it to reflect the paper better.

Line 17-18. I would advise the authors to transform this sentence into a more informative sentence. In other words, do not mention what you are discussing in the manuscript without adding something about it, but rather, be specific and give highlight details of the results and conclusions that can be made concerning this method. It would not necessarily need to make the text much longer, but all the more informative for future readers.

Line 21. In-text references, it is neither alphabetical nor chronologically, please pick something and be consistent throughout the manuscript.

Line 30. This is the wrong reference to Schwarz et al. and his work on rBC in snow. The 2006 paper deals with atmospheric BC. I would recommend the authors to look for his 2012 paper, that deals with rBC in snow samples.

Lines 46-48. It is difficult to understand what the authors mean with this sentence. How is it not the main point? (or do you simply mean that it is not the point in the paper to differentiate between WinsOC and OC?). How can the concept be applied to the analysis of both types?

Line 53. Adjusted the method accordingly how? Please provide the reader with what was done previously (even if only brief).

Lines 56-57. Please be more specific in this sentence. This is very broad and not very descriptive of what is addressed in the paper.

Line 65. It is stated that snow samples were collected near the GAW station. Please be more precise. Overall, any more info that could be provided on the snow samples could potentially be useful for future work. One suggestion would be an informative table (which could easily be put to the supplement) containing relevant info on the snow, e.g. fresh/old, density, sampling depth, etc.)

Line 76. Please define PM10.

Line 96. What is method 3052? Please inform (even in brief).

Lines 116-131. The text along these lines can easily by moved to an introductory (or background section), as these results are not specific to this study, but rather has been known from previous work.

Line 117. TOA of snow samples. Obviously, the actual snow sample is not analyzed in the TOA, but rather the filter with collected particulates from the snow. Please adjust this throughout the manuscript to be as concise as possible.

Line 118-119. Does it not depend on what type of minerals are present on the filter? Some minerals will not change color, nor darken the filter substrate. If there is hematite present on the filter it is evident that the transmittance is affected, but not all MD. It is stated further down in the manuscript (line 226) that hematite was a minor contributor to the MD. Please be specific in the text.

Line 132. How was the Fe content determined? And also, what do these numbers translate into for MD concentration for the snow samples? Please include this as a guide for future readers who might have a MD concentration and would want to compare that to the range of samples where the approach here is applicable to.

Line 150. What is meant with high background of the filter matrix? If it refers to hematite, how did it get into the background of the filter?

Lines 180-183. One could argue that this information is not novel to the study here, but has rather been known from previous work, and is actually part of the motivation for this work. Thus, it should be moved to the introduction/or alternatively in a background section after the introduction.

Line 199. Please remove the equation from the text, it should be inserted on a separate line (as well as any other question from the text).

Line 210-211. Unloaded filters at room temp $I_0$? Previously, it is stated that $I_0$ is for 400°C. Please clarify.

Lines 217-218. It is mentioned that such high hematite loadings are rarely found. What type of concentration does it equal in the snow? Please provide numbers on this, is it above ppm?

Line 225. SEM analyses confirmed such differences. Please be more specific by including some results.

Lines 225-226. Hematite was larger for the reference than the snow originating particles. Any idea of how much larger? Please include this information.

Fig. 4. The figure would benefit if both axes would be modified to more accurately reflect the range of the sampling points. In other words, the sampling points would be more centered in the figure.

Lines 278-279. How do we know of the different originating Fe, as in where is the evidence for this claim? Visual inspection also?

Line 285. Please add that it is during the calibration stage during TOA.

Line 290. Have the authors actually tested this for the other protocols? It should be the case, but if it was tested, this sort of statement could be made with greater confidence.

Data availability. This does not comply with current AMT standards on data availability.

---

## Referee Comment (RC2)

The authors present an approach which utilizes this interference to determine the concentration of hematite via thermal-optical analysis using a Lab OC/EC Aerosol Analyzer (Sunset Laboratory Inc.) and the EUSAAR2 protocol. Generally, the manuscript is well written and easy to follow. The investigation fits the scope of this journal. However, I think the major defect of this study is that only illustrates the Fe effect on EC measurement, but without any discussion about the detection of OC concentration. Therefore, I suggested that the manuscript can be accepted only with major revisions as follows.

Major issues:

1. The calibration of Fe in snow samples by the TOA method is quite useful. However, I am a little puzzled that why the authors are only interested in the occurrence of mineral dust in snow samples. Does it mean that this investigation is only useful for snow samples? As the author illustrated that the $PM_{10}$ samples can also obtain high amounts of Fe loadings, such as tunnel samples.

2. Reconstructed the abstract, as shown above, why only mineral dust and elemental carbon in snow samples are pained more attention? The mineral dust can also lead large bias of EC and OC concentrations for aerosol samples due to the temperature dependency of the transmittance signal determination.

3. I wonder to know what's the relationship between the attenuation and the hematite loading of less than 10 $\mu gFe\ cm^{-2}$. Because the bulk aerosols or snow samples can be loaded with lower Fe concentrations.

4. Although the major issue of this study focused on the bias of Fe on TOA techniques, However, there may be a large bias to account for insoluble OC concentration by using a microwave during the snow melt process.

5. In section 2.3, the author should provide more description of the relationship between the MD and $Fe_2O_3$, without only cited with previous studies.

6. Same as above section 3.1 is too general, the author should provide more details about the procedure of the treatment.

7. In section 3.2, as least, the detection limit of $Fe_2O_3$ should be given by ICP-MS or OES.

8. Actually, section 3.4 and Figure S2 are nothing useful and can be deleted directly.

9. The caption of Figure S2 is unclear. Does Figure S2 is standard samples or observed samples?

10. Finally, as Wang et al. (2012) indicated that the mineral dust mainly induces an extra decrease in optical reflectance during the 250 $^{O}$C heating stage, thereafter, lead potential bias in the EC and OC split. But I didn't find any related illustration or explanation of such an issue in this study. I suggested the author should provide further details on this major issue of the split of EC and OC in snow samples to prove this useful approach.

11. Finally, the author should note that there is potential mass loss of BC or MD on 1.0 μm quartz fiber filters compared with 0.4 μm Nuclepore filters, as shown in Figure 5 by Wang et al. (2020).

12.

References:

Wang M, Xu B, Zhao H, Cao J, Joswiak D, Wu G, et al. The Influence of Dust on Quantitative Measurements of Black Carbon in Ice and Snow when Using a Thermal Optical Method. Aerosol Science and Technology 2012; 46: 60-69.

Wang X, Zhang XY, Di WJ. Development of an improved two-sphere integration technique for quantifying black carbon concentrations in the atmosphere and seasonal snow. Atmospheric Measurement Techniques 2020; 13: 39-52.

---

## Author Response (AR1)

Reply to referee comment 1 on Kau et al. "Thermal-optical analysis of snow samples – challenges and perspectives introduced via the occurrence of mineral dust"

We thank Jonas Svensson for taking time to review our manuscript. The text of the referee comment is in **bold**, while our reply is in regular type. Passages quoted literally from the revised manuscript are in *italic* type.

**Referee report on Kau et al. 'Thermal-optical analysis of snow samples – challenges and perspectives introduced via the occurrence of mineral dust.'**

**General comments**

**This manuscript presents a new approach in dealing with the interference of mineral dust (hematite in particular) that might occur during thermal-optical analysis of filter substrates. The paper would be highly welcomed in the literature, due to the reoccurring phenomenon of mineral dust disturbing OCEC-analyzes. Since it is a novel approach, the text is well suited for AMT. In its current form, however, the paper needs some substantial work. Currently, the full potential of the manuscript has not been explored, but rather, corners appear to have been cut, resulting in several sections that either lack or provide minimal information. As an example, the approach is tested for high alpine PM10 filter samples, but only two samples are available. More data would be needed to provide a more robust basis for the authors to discuss the results on that topic. Is there data available data from another site (other than Sonnblick?). Similarly, the snow samples from Sonnblick could be compared with snow samples from another location. This would mean more work, but in the end, it would only and built-up the discussion/conclusions, and ultimately, strengthen the manuscript. With that said, I would reiterate that major revisions are necessary for this paper that could have great potential breakthrough in this area of science. Please see more comments below.**

We thank the referee for mentioning that the paper would be highly welcomed in literature and the valuable input to improve our manuscript. We addressed all of the requests mentioned as major or minor comments below and added new data to the manuscript.

To address the points mentioned above, within the general comments, we have two remarks:
1) A higher number of $PM_{10}$ filters is included in the revised version of the manuscript and another reference material was tested to set a more robust basis for our approach.
2) The only point we could not cope with is to add results of snow and $PM_{10}$ samples collected at different sites. Still, we are convinced that our data set of snow and $PM_{10}$ samples is comprehensive enough to justify the publication in its revised form. All samples containing mineral dust originate from Sonnblick, but they represent quite a variability as, regarding snow samples, they were collected during different visits at the site carried out during the years 2018 to 2020. Regarding $PM_{10}$ samples, the time period of 2017 until 2022 is covered. Thus, we are confident that we can present sound results.

Testing of the approach for different sites and thus environmental conditions is important and necessary but goes beyond the scope of a manuscript presenting a novel method utilizing TOA for the determination of iron loadings of mineral dust samples.

**Major comments**

**Section 3. The different instrumentation used in the study seem applicable and impressive. Still, it is difficult for a reader to see their place and role in the different steps of the approach. It could easily be misinterpreted that the instruments were just randomly chosen when the authors clearly thought about how to best utilize their instruments for this method. One suggestion is to provide an introductory section before each instrumentation is expanded upon. In this introductory text, it should be explained why the instrument is used. In a way, it should end up as a flow-chart explaining the different steps of the analysis, providing future readers with a roadmap of the approach enabling them to easier follow along.**

We agree that this contributes to a better understanding of our work. Consequently, we adapted all sub-sections of Section 3 (line 106 - 110, 117 - 119, 128, 138 - 139 in the revised manuscript). Now the reason for using these methods is stated before further explanations.

The different sections now read:

3.1 Thermal-optical analysis:

'*Thermal-optical analysis (TOA) is the reference method for the determination of OC and EC in ambient aerosols (DIN EN 16909:2017). Filter aliquots are heated in an inert and oxidizing atmosphere, the evolving carbonaceous compounds are converted to methane which is quantified via a flame ionization detector. To correct for pyrolytic carbon, the transmittance of the filter aliquot is recorded. As explained in detail in the results section, the development of the transmittance signal is influenced by the Fe loading of the filter. Therefore, we solely investigate this transmittance signal within this manuscript. A Lab OC-EC Aerosol Analyzer (Sunset Laboratory Inc., USA) and the EUSAAR2 protocol (Cavalli et al., 2010; temperature steps are given in Table S1 in the Supplement) were used. The instrument logs the transmittance and reflectance of the sample at a wavelength of 660 nm during the measurement. For the measurement, the program OCEC834 and for the evaluation of the raw data the program Calc415 (both Sunset Laboratory Inc., USA) were used. Further processing of the transmittance data was conducted with an external program (Microsoft Excel, Microsoft Corporation, USA).*'

3.2 Inductively coupled plasma-optical emission spectroscopy and -mass spectrometry:

'*Inductively coupled plasma-optical emission spectroscopy (ICP-OES) and inductively coupled plasma-mass spectrometry (ICP-MS) were used to quantify Fe. For this, a microwave assisted digestion of the filter aliquots was conducted subsequent to TOA. Filters loaded with snow samples or reference materials ($Fe_2O_3$, SRM 2709) samples and high alpine $PM_{10}$ samples were digested using a microwave system (Multiwave 5000, Anton Paar, Austria) and METHOD 3052 (method for microwave assisted acid digestion suitable for siliceous, organic and other complex matrices; US EPA, 1996; $HNO_3$:HCl:HF 6:2:1, maximum temperature: 180°C) and were analyzed for Fe via ICP-OES (iCAP 6500 ICP-OES spectrometer, Thermo Scientific, USA). Filter aliquots of the particulate matter samples collected within the railway tunnel were digested using a microwave system (Multiwave 3000, Anton Paar, Austria/Start1500, MLS GmbH, Germany; aqua regia, maximum temperature: 220°C) and Fe was quantified via ICP-MS (iCap Q System instrument, Thermo Scientific, USA). The limit of detection was 0.4 µg cm$^{-2}$ for ICP-OES and 0.1 µg cm$^{-2}$ for ICP-MS.*'

3.3 X-ray powder diffraction:

'*X-ray powder diffraction (PXRD) was used for specification of Fe present on the snow sample filters. Experiments were carried out in Bragg Brentano geometry using an Empyrean diffractometer (Malvern PANalaytical B.V., Netherlands; scattering angle range of 5° < 2$\theta$ < 135°). A focussing mirror was used to provide Cu $K_{\alpha1,2}$- radiation for the experiment. The beam divergency was defined by using a 1/4° fixed vertical entrance slit followed up by a 0.04 rad horizontal Soller slit and a 0.04 rad horizontal Soller slit on the secondary side in front of an open line detector (GaliPix detector). The detector to sample distance for this instrument was fixed to 240 mm.*

*The PXRD diagrams were evaluated using the Malvern PANalytical program suite HighScorePlus v4.6a (Degen et al., 2014). A background correction and a $K_{\alpha2}$ strip were performed. Crystallographic phases were assigned based on the ICDD-PDF4+ database (Kabekkodu et al., 2002).*'

3.4 Scanning electron microscopy:

'*Scanning electron microscopy (SEM) measurements allowed to visualize particle sizes and their elemental composition. Thus, a comparison of filters loaded with snow samples and the reference substances $Fe_2O_3$ and SRM 2709 was possible. A FEI Quanta 200 (Thermo Fisher Scientific, USA) instrument equipped with an Octane Pro EDS System (EDAX, USA) was used for the analyses of loaded quartz fiber filters. Samples were coated with Au prior to analysis.*'

**The aims of the paper should be scrutinized. Currently, there is a mismatch between the aims (denoted as interests in the introduction) stated in the introduction and once they are re-stated in the conclusions. The aims should be clearly reviewed and revised in the introduction. On the whole, the conclusions that are discussed particularly in lines 298-308 are not supported by the data currently presented in the paper (e.g. recommending a rerun of analyses; that the approach was successfully applied to PM10 filters also, see additional comment on this below).**

Thank you for this remark! To solve this mismatch, we harmonized the aims stated in the introduction (lines 68 - 72) and the conclusions (lines 342 - 362) in the revised manuscript).

Now they read:

Introduction:

'*In this work we investigate the influence of MD loads on TOA of snow samples collected in a high alpine environment. The main interest of our work is to investigate the temperature dependence of the light attenuation caused by Fe containing compounds in MD. This led us to a new approach to approximate Fe concentrations via TOA. Therefore, we evaluate the transmittance signal during the calibration phase, i.e. when carbonaceous compounds are no longer present. The method is tested extensively for snow samples and evaluated briefly for particulate matter samples.*'

Conclusion:

'*Investigating the influence of MD, i.e. its light absorbing constituent hematite, on TOA, we presented an easily applicable approach to determine Fe loadings on filters containing MD based on temperature changes of the transmittance signal recorded during the calibration phase of TOA. The method is based on the evaluation of quartz fiber filters loaded with snow samples containing different amounts of MD*

*Considering the whole range of Fe loadings investigated, the relationship between the Fe loading and $ATN_{700-400}$ was shown to be non-linear. By setting two straightforward criteria ($ATN_{700-400}$ between 3.5 and 29 and transmittance signal at 400°C above 600 a.u.) the determination of Fe loadings between 10 to 100 µgFe cm⁻² becomes possible based on a linear approach. This approach opens the possibility to obtain another compound of LASI with the same method (TOA) as used for the determination of EC. This can be advantageous if the available amount of sample is limited and a proxy for MD is needed. Still a reliable conversion factor of Fe to MD has to be deduced by independent analysis or taken from literature and is not presented here.*

*The method developed for the analyses of filters loaded with snow samples was successfully applied to the analyses of particulate matter filters originating from the same high alpine environment. We cannot give information about the general applicability to PM10 samples loaded with different types of MD, yet. We hypothesize that it should be possible to use the approach as long as MD and its constituent $Fe_2O_3$ remain the dominant source of Fe loadings. Still, an independent calibration might be needed to account for differences in MD composition. Fe loadings in particulate matter samples collected within a railway tunnel and thus having a strong influence of abrasion products from tracks and wheels and not from MD cannot be analyzed with the method presented here. This illustrates the limitation of the method.*

*Filters loaded heavily with MD (Fe loadings above 10 µgFe cm⁻²) will experience a severe bias of the OC/EC split point. EC will be underestimated or cannot be determined at all, while OC will be overestimated.*

*We recommend to rerun the analyses, as soon as the evaluation of the transmittance signal in the calibration phase indicates an influence of MD. This second run of analyses will allow to set the split point more precisely.*'

The recommendation to rerun TOA is based on the data of repeated analyses presented in Section 4.1. and is actually already mentioned there (lines 178 - 179, 188 - 189 in the original manuscript). The topic regarding PM$_{10}$ filters is discussed below.

**Section 4.2 and Fig.2. For the samples contaminated with the reference hematite, was it one batch of filter samples? In other words, was this step ever repeated, with another independent set of hematite contaminated filters? In order to test whether the same pattern would be repeated in the filter samples this is a suggestion of work to be done. Similarly, what about conducting experiments with reference filters containing Fe originating from a different source than hematite? This could further confirm the claims made that this method works well for hematite by not other types (which currently is not strongly supported by much data; only the tunnel samples).**

The data presented in Figure 2 refers to one batch of filter samples, i.e. a series of 9 filters loaded with different amounts of the same reference sample, i.e. a suspension of $Fe_2O_3$ in water. At a different time two additional filters were loaded with a suspension prepared independently. These two data points (8.8 and 19 µgFe cm⁻²) fit perfectly to the existing data and are included in Figure 2 and Figure 5a in the revised manuscript. Furthermore, we mention that two suspensions were prepared independently. Thus, we can confirm that the same pattern is obtained when repeating this part of analysis.

Still, we agree with the comment that experiments with reference filters containing Fe from different sources would be interesting, though not absolutely necessary at this stage.

First, we want to explain why we chose hematite. Formenti et al. (2014) present the mineralogical composition of mineral dust in Western Africa. Fe oxides (mainly hematite and

goethite) account only for a low percentage of MD mass (2 - 5 %), but they represent the biggest group of Fe compounds in MD (roughly 58 %). At elevated temperatures (250 - 600°C), goethite changes to hematite (Liu et al., 2013). This temperature is exceeded during TOA already in the inert phase. Therefore, hematite will be the main Fe compound on our sample filters during the oxygen phase and the calibration phase and it is well suited as a reference material for our method. As the occurrence of hematite in the snow samples was confirmed in the PXRD analyses, we focused on this compound rather than addressing different isolated compounds. Actually, we checked for FeO and this compound did not show changes in transmittance when heating or cooling the samples. Still, FeO was selected rather arbitrarily, and we did not include a respective paragraph in the text as we do not have any indication that FeO is relevant for snow or PM samples containing MD. The information why we use hematite as a reference material is added in the revised manuscript in the Introduction and when the reference samples are introduced (Introduction: line 57 - 63, Section 2.3: lines 94 - 97).

Testing single minerals will give a 'yes/no' decision whether an effect is possible and are definitely valuable. Still the conditions found for actual samples will be different, as other minerals contained in MD and particle sizes will influence the transmittance as well due to multiple scattering and different loading effects. To go in that direction, an independent set of filters loaded with standard reference material SRM 2709 (San Joaquin Soil, NIST) was prepared and analyzed using TOA, ICP-OES and SEM. SRM 2709 is an agricultural soil, intended to be used as a reference for soil or materials of similar matrix. Unfortunately, we do not have a specification of iron contained in this SRM. Our evaluations show that the same patterns are visible for different sample groups, e.g. high alpine samples, SRM 2709 or hematite, but different fits are necessary to relate Fe and $ATN_{700-400}$. The reference substances serve the mere purpose to show that their overall trend of $ATN_{700-400}$ vs. Fe loading is similar to the trend observed for the actual snow sample filters.

The data about SRM 2709 was added in line 101 - 103, 242 - 244, 256 – 262, Figure 2 and Figure 5a in the revised manuscript.

Section 2.3:

'*A different set of samples was prepared with Standard Reference Material® 2709 (SRM 2709 San Joaquin Soil, National Institute of Standards and Technology, USA). SRM 2709 was suspended in ultrapure water (83.3 mg in 1 L flask) and quartz fiber filters were loaded covering the range of roughly 3 to 141 µgFe cm$^{-2}$.*'

Section 4.2:

'*First, we evaluate the transmittance signal of filters loaded with two reference substances, $Fe_2O_3$ and SRM 2709.*'

'*We attribute the differences between the filters loaded with SRM 2709 and $Fe_2O_3$ shown in Fig. 2 to differences in the overall loadings of the filters (i.e. the loading with other material than hematite) and the particle sizes. Such differences are confirmed via SEM analyses of selected filters loaded with both reference materials. While $Fe_2O_3$ filters only contained hematite, SRM filers showed different kinds of minerals, i.e. particles consist of Ca, K, Si, Al, Mg, Na and O and just a minor contribution of Fe, which obviously leads to lower ATN values. Hematite particles on the $Fe_2O_3$ filters were generally larger in size compared to Fe-rich particulate matter in SRM 2709. Examples of the SEM analyses, elemental maps and spectra are given in the Supplement (Fig. S3).*'

[Figure]

**Figure 1: Fe loading of Fe$_2$O$_3$ samples determined via ICP-OES plotted versus ATN$_{700-400}$.**

The group of tunnel samples, where Fe is present due to abrasion of tracks and wheels and therefore is present in other forms than hematite as well (mainly magnetite and metallic Fe) show that the applicability of our approach is limited. This set of samples is provided as some kind of warning, that TOA cannot be applied to determine Fe in all sample matrices. We tried to be more specific about this point in the Conclusions:

'*The method developed for the analyses of filters loaded with snow samples was successfully applied to the analyses of particulate matter filters originating from the same high alpine environment. We cannot give information about the general applicability to PM$_{10}$ samples loaded with different types of MD, yet. We hypothesize that it should be possible to use the approach as long as MD and its constituent Fe$_2$O$_3$ remain the dominant source of Fe loadings. Still, an independent calibration might be needed to account for differences in MD composition. Fe loadings in particulate matter samples collected within a railway tunnel and thus having a strong influence of abrasion products from tracks and wheels and not from MD cannot be analyzed with the method presented here. This illustrates the limitation of the method.*'

**Section 4.3. It is understood that there are only two PM$_{10}$ samples from Sonnblick. Yet, two samples are not enough to support the claims the authors want to make in the text (concerning the applicability of the fit for PM$_{10}$ samples). The authors need to find a way to expand with more samples for the PM$_{10}$ or majorly down-play the importance of only two samples in the claims they are making.**

We agree that the low number of PM$_{10}$ samples from Sonnblick was unfortunate. As stated in line 245, loadings above 10 µgFe cm$^{-2}$ were hardly achieved during the long-term data set covering more than 4 years.

Due to the frequent and strong occurrence of MD this spring (the data became available just after the manuscript was submitted) the number of PM$_{10}$ filters could now be increased from 2 to 6. On the one hand this is an increase by 200 %, on the other hand one could still argue that a data set of six samples is not huge. The data set is too small to calculate a fit for PM$_{10}$ samples themselves but shows that the data of PM$_{10}$ samples collected at the same environment fits nicely into the fit obtained for snow samples. Hence, we only present the data compared to the snow samples and calculate differences between the Fe concentration using the ICP-OES data and our fit for the snow samples and do not calculate another fit for the PM$_{10}$ samples.

The PM$_{10}$ dataset was enlarged, and the new data added to Figure 4 and Figure 5a. To account for the additional samples, the text was adapted in line 292 - 297 in the revised manuscript:

'*Only six samples out of the data set showed Fe loadings in the appropriate range. The PM$_{10}$ samples represent events with long range transport of desert dust to the high alpine site and are in good agreement with the model obtained for the snow sample filters. These data points are also included in Fig. 4. Applying the fit calculated for the snow samples, the relative differences between the Fe loadings determined with TOA and with ICP-OES were between 3% and 49% with a median of 24% for the PM$_{10}$ samples.*'

[Figure]

**Figure 2: Linear fit for Fe loading and ATN$_{700\text{-}400}$ in snow samples and comparison with high alpine PM$_{10}$ samples showing sufficiently high Fe loadings.**

**Minor comments**

**Title. As it currently reads the title of the paper suggests that *challenges* and *perspectives* are to be discussed. The use of such wording would be more appropriate if this was a review article. I would encourage the authors to critically think through the title and modify it to reflect the paper better.**

We agree and changed the title to '*Thermal-optical analysis of quartz fiber filters loaded with snow samples - determination of iron based on interferences caused by mineral dust*'.

**Line 17-18. I would advise the authors to transform this sentence into a more informative sentence. In other words, do not mention what you are discussing in the manuscript without adding something about it, but rather, be specific and give highlight details of the results and conclusions that can be made concerning this method. It would not necessarily need to make the text much longer, but all the more informative for future readers.**

Thank you for this input, we absolutely agree. The last sentence was rewritten (line 18-23 in the revised manuscript):

'*The method, initially designed for snow samples, can also be used to evaluate particulate matter samples collected within the same high alpine environment. When applying the method to a new set of samples it is crucial to check whether the composition of iron compounds and the sample matrix remain comparable. If other sources than mineral dust determine the iron concentration in particulate matter, these samples cannot be evaluated with thermal-optical*

*analysis. This is shown exemplarily with data of particulate matter samples collected in a railway tunnel.'*

**Line 21. In-text references, it is neither alphabetical nor chronologically, please pick something and be consistent throughout the manuscript.**

We listed in-text references alphabetically throughout the manuscript (e.g. line 26 in the revised manuscript).

**Line 30. This is the wrong reference to Schwarz et al. and his work on rBC in snow. The 2006 paper deals with atmospheric BC. I would recommend the authors to look for his 2012 paper, that deals with rBC in snow samples.**

Thank you for pointing this out. We changed the reference accordingly (line 35 and line 466 - 469 in the revised manuscript).

**Lines 46-48. It is difficult to understand what the authors mean with this sentence. How is it not the main point? (or do you simply mean that it is not the point in the paper to differentiate between WinsOC and OC?). How can the concept be applied to the analysis of both types?**

Yes, we mean that it is not the point in the manuscript to differentiate between WinsOC and OC. Depending on the kind of sample, TOA yields either WinsOC (analysis of insoluble particles in e.g. snow filtrated onto quartz filters) filters or OC (analysis of ambient particulate matter collected on quartz filters). However, there is no analytical difference between WinsOC and OC, therefore the evaluation we describe can be applied to both sample types. For clarification, we avoided the unclear sentence and adjusted the text in line 51 - 53 in the revised manuscript:

'*As we focus on the analytical procedure of TOA, we will avoid the term WinsOC throughout the manuscript and stick to OC. The concept of analyses and evaluation can be applied to the analysis of both sample types, i.e. particulate matter filters (then OC is addressed) or filters loaded with liquid snow samples (then WinsOC is addressed).'*

**Line 53. Adjusted the method accordingly how? Please provide the reader with what was done previously (even if only brief).**

Wang et al. (2012) shifted the reference value for the OC/EC split to the 250°C temperature step. We added this information in line 66 in the revised manuscript. The sentence now reads:

'*Wang et al. (2012) used TOA for analyses and identified an extra decrease in optical reflectance during the 250°C heating stage in hematite and suggested to shift the reference value for the OC/EC split to this temperature step.'*

**Lines 56-57. Please be more specific in this sentence. This is very broad and not very descriptive of what is addressed in the paper.**

We revised the sentence to make it more informative (line 68 in the revised manuscript):

'*In this work we investigate the influence of MD loads on TOA of snow samples collected in a high alpine environment.*'

**Line 65. It is stated that snow samples were collected near the GAW station. Please be more precise. Overall, any more info that could be provided on the snow samples could potentially be useful for future work. One suggestion would be an informative table (which could easily be put to the supplement) containing relevant info on the snow, e.g. fresh/old, density, sampling depth, etc.)**

The samples were collected on the upper platform of the observatory. Regarding sampling depth, the uppermost 5 centimeters of snow were collected. We added this information in line 77 in the revised manuscript. As these samples were collected for method development, no further information, e.g. density, time of last snowfall, was recorded.

**Line 76. Please define PM10.**

We added the definition of $PM_{10}$ in line 88 in the revised manuscript:

'*$PM_{10}$, i.e. particles with an aerodynamic diameter equal to or less than 10 μm, was sampled onto quartz fiber filters (Pallflex® Tissuquartz™) in the same high alpine environment as the snow samples using a high volume sampler (DIGITEL Elektronik AG, Switzerland; sampling duration: 1 week) and in tunnels with railway traffic using a low volume sampler SEQ47/50 (Sven Leckel Ingenieurbüro GmbH, Germany; sampling duration: 4 h).*'

**Line 96. What is method 3052? Please inform (even in brief).**

We added a brief description of METHOD 3052 in line 120 - 122 in the revised manuscript:

'*Filters loaded with snow samples or reference materials ($Fe_2O_3$, SRM 2709) samples and high alpine $PM_{10}$ samples were digested using a microwave system (Multiwave 5000, Anton Paar, Austria) and METHOD 3052 (method for microwave assisted acid digestion suitable for siliceous, organic and other complex matrices; US EPA, 1996; HNO3:HCl:HF 6:2:1, maximum temperature: 180°C) and were analyzed for Fe via ICP-OES (iCAP 6500 ICP-OES spectrometer, Thermo Scientific, USA).*'

**Lines 116-131. The text along these lines can easily by moved to an introductory (or background section), as these results are not specific to this study, but rather has been known from previous work.**

Only the text in lines 117 - 120 is general knowledge. The following sentences are explanations of Figure 1 and definitely needed in section 4.1. Still, we think it is important to keep the first three sentences so that the following arguments and figures are well understood, especially for people not being absolutely familiar with TOA.

**Line 117. TOA of snow samples. Obviously, the actual snow sample is not analyzed in the TOA, but rather the filter with collected particulates from the snow. Please adjust this throughout the manuscript to be as concise as possible.**

We agree that we were not precise enough. We checked the whole manuscript and adjusted the text accordingly.

**Line 118-119. Does it not depend on what type of minerals are present on the filter? Some minerals will not change color, nor darken the filter substrate. If there is hematite present on the filter it is evident that the transmittance is affected, but not all MD. It is stated further down in the manuscript (line 226) that hematite was a minor contributor to the MD. Please be specific in the text.**

We agree that MD can vary in its composition. MD deposited on Mount Sonnblick often derives from long range transport of dust from desert regions (as given in lines 66 – 67 in the original manuscript). But even for desert regions several differences exist.

Fe oxides usually come up to a few percent of MD. This was already explained further up (major comments, Hematite as a reference compound). The composition of MD and the special situation of hematite is discussed in more detail in the revised version of the manuscript as indicated above.

Furthermore, we also revised the sentences addressed in this comment. If MD contains reddish Fe compounds, as hematite, the filter will remain colored after TOA. We added this information in line 146 in the revised manuscript to be more precise.

Still, this comment made us to recheck the wording in the whole manuscript. Now we only refer to hematite when the actual reference sample is addressed. We changed the description in the snow and $PM_{10}$ samples from hematite to an expression like 'Fe containing compounds'. Fe oxides described to be the important light absorbing compound in MD (Alfaro et al., 2014); however, the analysis of Fe via digestion and ICP-OES or ICP-MS leads to the Fe concentration of all accessible Fe compounds including Fe in other forms, if present. Hence, the description similar to "Fe containing compounds" is more precise, especially if samples from different origins with varying composition are compared. This also accounts better for the fact that differences between our samples from Sonnblick and the SRM filters – loaded with different kinds of dust – are visible.

**Line 132. How was the Fe content determined? And also, what do these numbers translate into for MD concentration for the snow samples? Please include this as a guide for future readers who might have a MD concentration and would want to compare that to the range of samples where the approach here is applicable to.**

Fe was determined using ICP-OES or ICP-MS for all samples (line 94-101).

We do not want to give MD concentrations, as it is difficult to provide a robust conversion factor from Fe to MD (see the section about the composition of MD and the references to Kandler et al., 2007 and Formenti et al., 2014). Still, we can provide Fe concentrations of the two snow samples mentioned in line 132. They were as follows: the sample containing MD had an Fe concentration of 1.1 mgFe $L^{-1}$ (filtrated volume 54 mL), while for the sample without visible MD contamination the Fe concentration was 51 µgFe $L^{-1}$ (filtrated volume 99 mL). This is well within the concentration range given in literature, e.g. by Kaspari et al. (2014), who report Fe concentrations in snow and ice ranging between <10 µgFe $L^{-1}$ and 100 mgFe $L^{-1}$.

We added the Fe concentration in the liquid snow sample and the filtrated volume of liquid snow in line 160 - 162 in the revised manuscript.

**Line 150. What is meant with high background of the filter matrix? If it refers to hematite, how did it get into the background of the filter?**

The high background is due to the quartz fiber filters which need to be used for TOA.

**Lines 180-183. One could argue that this information is not novel to the study here, but has rather been known from previous work, and is actually part of the motivation for this work. Thus, it should be moved to the introduction/or alternatively in a background section after the introduction.**

We agree that the description of the conditions during the calibration phase is quite general, but we are not aware of literature where the transmittance signal during the calibration phase is utilized. Thus, it is novel. As we feel that this passage adds to understanding why this approach is suitable (no interference from carbonaceous compounds, oxidizing conditions as prevailing when the OC/EC split is usually set) we decided to leave it in the Results section.

**Line 199. Please remove the equation from the text, it should be inserted on a separate line (as well as any other question from the text).**

Thank you for making us aware of this. We removed the equations in lines 199 and 241 from the text and added cross references in the text.

**Line 210-211. Unloaded filters at room temp $I_0$? Previously, it is stated that $I_0$ is for 400°C. Please clarify.**

We defined the transmittance at 400°C as $I_0'$ (line 203 – 206) (note the apostrophe). $I_0$ is the laser transmittance at the end of TOA reflecting an unloaded filter (line 199 – 201), we used $I_0$ for the transmittance value of an unloaded quartz fiber filter at room temperature.

**Lines 217-218. It is mentioned that such high hematite loadings are rarely found. What type of concentration does it equal in the snow? Please provide numbers on this, is it above ppm?**

The wording was misleading. Such high loading can be found (you just need to filtrate larger amounts onto a small filter). The reason for such high loadings being rarely found is that they are difficult to handle. The filter will be overloaded, which leads to slow filtration and eventually to loss of the deposited material during handling. To clarify this, we changed to sentence which now reads '*Fe loadings above 150 µgFe cm$^{-2}$ are avoided, as these filters are prone to sample loss*'. (line 254 - 255 in the revised manuscript).

To provide numbers, we added a short calculation: The sample with visible occurrence of MD presented in section 4.1 contained 1.1 mgFe L$^{-1}$. To reach the upper part of the curve, i.e. an Fe loading above 150 µgFe cm$^{-2}$, roughly 270 mL need to be filtrated. Visible occurrence means that MD was visible within the snow and after melting.

**Line 225. SEM analyses confirmed such differences. Please be more specific by including some results.**

We included more information about SEM analyses in the revised text (lines 256 - 262, 269 - 271) and in the Supplement. We added elemental maps of Fe for a filter loaded with SRM 2709 and particles of a melted snow sample. The elemental maps show that Fe-rich particulate matter on SRM and snow filters are generally smaller than particles of $Fe_2O_3$ reference filters. Additionally, we added spectra of a filter loaded with $Fe_2O_3$, SRM 2709 and a snow filter. These spectra show signals only for Fe, O, the quartz fiber filter and the Au coating for $Fe_2O_3$, and signals for Fe, Ca, K, Si, Al, Mg, Na and O as well as the Au coating for the SRM and snow filter. The elemental maps and spectra are part of Figure S2, while the description of Figure S2 and the text belonging to Figure S2 (now Figure S3 in the revised manuscript) were adjusted (line 40 - 51 in the revised Supplement).

Text of Figure S3:

'*The images (a), (b) and (d), recorded using backscattered electrons, show the atomic number contrast. Due to Fe's high atomic number, particles containing Fe are shown in a brighter color than the filter material or particles with different composition. Furthermore, they present an indication of particle sizes and number concentrations. The elemental maps of the filters loaded with SRM 2709 (c) and with particles of a liquid snow sample (e) show areas with high Fe concentrations colored in red. The EDS spectrum for the filter loaded with $Fe_2O_3$ (f) shows only signals of Si, O, Fe and Au. Besides $Fe_2O_3$, the filter material, $SiO_2$, is visible in the spectrum as well as the coating of the sample, Au. The EDS spectra of the filter loaded with SRM 2709 (g) and particles of a liquid snow sample (h) show signals of Ca, K, Si, Al, MG, Na, O, Fe and Au and thus underline the occurrence of different minerals.*'

**Lines 225-226. Hematite was larger for the reference than the snow originating particles. Any idea of how much larger? Please include this information.**

The sizes were determined only for a few particles. No size distribution was determined. A qualitative impression of particle sizes can be obtained from SEM images; however, we cannot provide reliable numbers for particle sizes.

We adapted the text regarding SEM analyses in line 256 - 262, 269 - 271 in the revised manuscript:

'*We attribute the differences between the filters loaded with SRM 2709 and $Fe_2O_3$ shown in Fig. 2 to differences in the overall loadings of the filters (i.e. the loading with other material than hematite) and the particle sizes. Such differences are confirmed via SEM analyses of selected filters loaded with both reference materials. While $Fe_2O_3$ filters only contained hematite, SRM filers showed different kinds of minerals, i.e. particles consist of Ca, K, Si, Al, Mg, Na and O and just a minor contribution of Fe, which obviously leads to lower ATN values. Hematite particles on the $Fe_2O_3$ filters were generally larger in size compared to Fe-rich particulate matter in SRM 2709. Examples of the SEM analyses, elemental maps and spectra are given in the Supplement (Fig. S3).*'

'*The chemical composition of particulates in snow is more similar to SRM 2709 than to pure hematite, as shown in Fig. S3 in the Supplement. Still, SEM images show differences in particle size of SRM and snow filters, which might induce the differences together with differences in the chemical composition, which cannot be identified via SEM.*'

**Fig. 4. The figure would benefit if both axes would be modified to more accurately reflect the range of the sampling points. In other words, the sampling points would be more centered in the figure.**

We edited Figure 4 accordingly:

[Figure]

Figure 3: Linear fit for Fe loading and $ATN_{700-400}$ in snow samples and comparison with high alpine $PM_{10}$ samples showing sufficiently high Fe loadings.

**Lines 278-279. How do we know of the different originating Fe, as in where is the evidence for this claim? Visual inspection also?**

While Fe in the snow samples originates from the mixture of minerals contained in MD, Fe in the tunnel samples originates mainly from abrasion of tracks, wheels and brakes of the trains. This was already reported at a previous comment. Eom et al. (2013) found that particulate matter samples from railway tunnels contained Fe in the form of magnetite, hematite, and metallic Fe (roughly 40%, 30% and 25%, respectively). Besides the overall iron loading of those filters being high, the relative contribution of iron to overall mass in tunnel samples is much higher (up to 73%, line 265) than for MD samples which contain 2 – 5% iron (Formenti et al., 2014). This is another reason why it cannot be MD on these samples. Additionally, visual inspection of the tunnel filters after TOA confirmed differences in the remaining compounds compared to the filters loaded with snow samples, as shown in Figure S4 in the Supplement.

The text was revised to highlight these facts in more detail (lines 310 - 315 in the revised manuscript):

'*To test whether the approach is generally applicable for aerosol samples with marked contributions of Fe compounds, we included samples collected within a railway tunnel. This represents a completely different environment as very high Fe contributions and different Fe compounds, like magnetite, hematite and Fe metal, can be expected (Eom et al., 2013), originating from abrasion of tracks, wheels and brakes of the trains. For the tunnel samples included in Fig. 5a, Fe accounts for up to 73% of particulate matter mass, but no specification of Fe compounds has been performed. This high share of Fe in particulate matter mass highlights the different origin of Fe compared to the high alpine samples, as Fe accounts for less than 5% of mass in MD only.*'

**Line 285. Please add that it is during the calibration stage during TOA.**

We added this information in line 344 in the revised manuscript.

**Line 290. Have the authors actually tested this for the other protocols? It should be the case, but if it was tested, this sort of statement could be made with greater confidence.**

Yes, few samples were analyzed using IMPROVE_A or NIOSH870 as well. As the results did not differ from the $ATN_{700\text{-}400}$ values obtained using EUSAAR2, we did not include this data to avoid complicating the explanation unnecessarily. We added a remark that the applicability of other protocols than EUSAAR2 was observed for a limited set of samples. We further moved the respective passage to Section 4.3 (line 335 - 340 in the revised manuscript), as it represents a result.

**Data availability. This does not comply with current AMT standards on data availability.**

Thank you for pointing this out. We added data of our samples ($ATN_{700\text{-}400}$ and Fe loadings) in Table S2 in the Supplement.

References:

Alfaro, S. C., Gomes, L., Rajot, J. L., Lafon, S., Gaudichet, A., Chatenet, B., ... & Zhang, X. Y. (2003). Chemical and optical characterization of aerosols measured in spring 2002 at the ACE-Asia supersite, Zhenbeitai, China. Journal of Geophysical Research: Atmospheres, 108(D23).

Eom, H. J., Jung, H. J., Sobanska, S., Chung, S. G., Son, Y. S., Kim, J. C., ... & Ro, C. U. (2013). Iron speciation of airborne subway particles by the combined use of energy dispersive electron probe X-ray microanalysis and Raman microspectrometry. Analytical chemistry, 85(21), 10424-10431.

Formenti, P., Caquineau, S., Desboeufs, K., Klaver, A., Chevaillier, S., Journet, E., & Rajot, J. L. (2014). Mapping the physico-chemical properties of mineral dust in western Africa: mineralogical composition. Atmospheric Chemistry and Physics, 14(19), 10663-10686.

Kandler, K., Benker, N., Bundke, U., Cuevas, E., Ebert, M., Knippertz, P., ... & Weinbruch, S. (2007). Chemical composition and complex refractive index of Saharan Mineral Dust at Izaña, Tenerife (Spain) derived by electron microscopy. Atmospheric Environment, 41(37), 8058-8074.

Kaspari, S., Painter, T. H., Gysel, M., Skiles, S. M., & Schwikowski, M. (2014). Seasonal and elevational variations of black carbon and dust in snow and ice in the Solu-Khumbu, Nepal and estimated radiative forcings. Atmospheric chemistry and physics, 14(15), 8089-8103.

Liu, H., Chen, T., Zou, X., Qing, C., & Frost, R. L. (2013). Thermal treatment of natural goethite: Thermal transformation and physical properties. Thermochimica Acta, 568, 115-121.

Wang, M., Xu, B., Zhao, H., Cao, J., Joswiak, D., Wu, G., & Lin, S. (2012). The influence of dust on quantitative measurements of black carbon in ice and snow when using a thermal optical method. Aerosol Science and Technology, 46(1), 60-69.

Reply to referee comment 2 on Kau et al. "Thermal-optical analysis of snow samples – challenges and perspectives introduced via the occurrence of mineral dust"

We thank X. Wang for taking time to review our manuscript. The text of the referee comment is in **bold**, while our reply is in regular type. Passages quoted literally from the revised manuscript are in *italic* type.

**Comment on amt-2022-145**
**X. Wang (Referee)**

**Referee comment on "Thermal-optical analysis of snow samples – challenges and perspectives introduced via the occurrence of mineral dust" by Daniela Kau et al., Atmos. Meas. Tech. Discuss., https://doi.org/10.5194/amt-2022-145-RC2, 2022**

**The authors present an approach which utilizes this interference to determine the concentration of hematite via thermal-optical analysis using a Lab OC/EC Aerosol Analyzer (Sunset Laboratory Inc.) and the EUSAAR2 protocol. Generally, the manuscript is well written and easy to follow. The investigation fits the scope of this journal. However, I think the major defect of this study is that only illustrates the Fe effect on EC measurement, but without any discussion about the detection of OC concentration. Therefore, I suggested that the manuscript can be accepted only with major revisions as follows.**

We thank the referee for the positive evaluation. We agree that the bias of the split point introduced by MD affects both EC and OC concentrations. Quite often OC concentrations are much higher than EC concentrations. Hence, variations of the OC/EC split have, on a relative base, a great impact on the EC concentrations. This is why we had our focus on EC. As we agree that MD has an impact on the OC concentrations as well, we revised the manuscript to address this problem in more detail. The respective changes are listed below, when addressing the referee's comments. Thus, we are confident that we could resolve the major point of criticism.

**Major issues:**

- **The calibration of Fe in snow samples by the TOA method is quite useful. However, I am a little puzzled that why the authors are only interested in the occurrence of mineral dust in snow samples. Does it mean that this investigation is only useful for snow samples? As the author illustrated that the PM$_{10}$ samples can also obtain high amounts of Fe loadings, such as tunnel samples.**

We absolutely agree, aerosol samples are affected as well, and this is why we included PM$_{10}$ samples collected at the background site to show the applicability of the method for PM$_{10}$ samples. The samples collected in a railway tunnel show the limitations of the method, as the TOA method failed to approximate iron loadings. Particles collected in the railway tunnel derive to a large extent from abrasion of tracks and wheels and iron is present in different forms than in mineral dust samples.

The importance of the method for analyzing particulate matter samples is further emphasized by increasing the number of PM$_{10}$ samples collected at the high alpine site with recent measurements (line 292 - 296, Figure 4 and Figure 5a). Furthermore, the Conclusions were revised to address this point further:

Conclusion:

'*We cannot give information about the general applicability to $PM_{10}$ samples loaded with different types of MD, yet. We hypothesize that it should be possible to use the approach as long as MD and its constituent $Fe_2O_3$ remain the dominant source of Fe loadings. Still, an independent calibration might be needed to account for differences in MD composition. Fe loadings in particulate matter samples collected within a railway tunnel and thus having a strong influence of abrasion products from tracks and wheels and not from MD cannot be analyzed with the method presented here. This illustrates the limitation of the method.*'

- **Reconstructed the abstract, as shown above, why only mineral dust and elemental carbon in snow samples are pained more attention? The mineral dust can also lead large bias of EC and OC concentrations for aerosol samples due to the temperature dependency of the transmittance signal determination.**

The abstract was changed to address both EC and OC and to refer to particulate matter samples more extensively (line 11 and 18 in the revised manuscript):

'*Still, the occurrence of mineral dust, which contains hematite, leads to a bias in the quantification of elemental carbon and organic carbon via thermal-optical analysis.*'

Other passages in the manuscript, where we originally just referred to EC, were changed as well and both EC and OC are mentioned now.

- **I wonder to know what's the relationship between the attenuation and the hematite loading of less than 10 µgFe cm$_{-2}$. Because the bulk aerosols or snow samples can be loaded with lower Fe concentrations.**

Figure 3 includes all data points of filters loaded with snow samples, i.e. also filters with loadings below 10 µgFe cm$^{-2}$. Filters with such low loadings show a small value for $ATN_{700-400}$ or even negative values (line 235 – 238). Thus, the scatter gets quite broad and no correlation is possible any longer. Figure 5(a) includes also high alpine $PM_{10}$ samples with loadings below 10 µgFe cm$^{-2}$ ($ATN_{700-400}$ close to 0).

- **Although the major issue of this study focused on the bias of Fe on TOA techniques, However, there may be a large bias to account for insoluble OC concentration by using a microwave during the snow melt process.**

The snow melt process is a delicate procedure. Concerns are expressed in literature that OC is lost when using a microwave or that, when melting the sample at room temperature, EC is lost on the walls of plastic bags and glassware. Evaluating these two sides, we decided to use the melting process via microwave, which is, according to Wang et al. (2020), a widely performed snow-melting procedure.

Authors applying our method in the future can still either use the microwave for melting or any other appropriate method.

- **In section 2.3, the author should provide more description of the relationship between the MD and Fe₂O₃, withoutonly cited with previous studies.**

The topic of taking hematite as a reference substance is explained in more detail in the revised version of the manuscript. MD is a complex mixture of various compounds. Fe oxides (hematite, goethite) in MD account for 2 to 5 % of MD mass and approximately 58 % of the mass of elemental Fe (Formenti et al., 2014). The exclusive use of hematite as a reference substance is appropriate as goethite changes to hematite at elevated temperature (250 – 600°C; Liu et al., 2013), which is exceeded during the inert phase of TOA. Thus, hematite will be the main Fe compound on the sample filters loaded with MD when TOA switches to the oxygen phase, i.e. at times when the split point is set, and during the calibration phase. This information is now added in the revised manuscript in the Introduction already (line 57 - 63) and we refer to this again in section 2.3 (lines 94 to 97):

Introduction:

'*Here we focus on iron oxides, which are the main light absorbing minerals in MD and cause its reddish (hematite) or yellowish (goethite) color (Alfaro et al., 2004). Formenti et al. (2014) found Fe oxides (goethite and hematite) to account for 2-5% of MD by mass, investigating the mineralogical composition of mineral dust in western Africa. At elevated temperatures, between 250 and 600°C, goethite changes to hematite (Liu et al., 2013). This temperature is exceeded already during the first part of TOA (helium atmosphere), making hematite the main Fe oxide present on the filters during the time when the split point for OC/EC differentiation is set. Thus, iron oxides do not only determine the properties of MD as a prominent component of LASI, but they also affect TOA.*'

Section 2.3:

'*Hematite (Fe₂O₃) was chosen as a reference substance for light absorbing compounds in MD as goethite and hematite are the most abundant forms of Fe oxides in MD (Formenti et al., 2014) and goethite will be converted to hematite at elevated temperatures starting at 250°C and being completed at 600°C (Liu et al., 2013), i.e. at temperatures which correspond to the conditions used in the inert phase of TOA already.*'

To evaluate our own data, PXRD of filters loaded with snow samples was conducted and showed Fe₂O₃ on the filters. The results of these measurements are presented in lines 148 - 149 (section 4.1.) in the original manuscript.

- **Same as above section 3.1 is too general, the author should provide more details about the procedure of the treatment.**

We added some details about the method (line 106 – 109 in the revised manuscript) and included a table in the Supplement defining the temperature steps of the EUSAAR2 protocol (Table S1 in the revised Supplement) and mention this in line 112 in the revised manuscript.

The text about thermal-optical analysis now reads:

'*Thermal-optical analysis (TOA) is the reference method for the determination of OC and EC in ambient aerosols (DIN EN 16909:2017). Filter aliquots are heated in an inert and oxidizing atmosphere, the evolving carbonaceous compounds are converted to methane which is quantified via a flame ionization detector. To correct for pyrolytic carbon, the transmittance of the filter aliquot is recorded. As explained in detail in the results section, the development of the transmittance signal is influenced by the Fe loading of the filter. Therefore, we solely investigate this transmittance signal within this manuscript. A Lab OC-EC Aerosol Analyzer (Sunset Laboratory Inc., USA) and the EUSAAR2 protocol (Cavalli et al., 2010; temperature*

*steps are given in Table S1 in the Supplement) were used. The instrument logs the transmittance and reflectance of the sample at a wavelength of 660 nm during the measurement. For the measurement, the program OCEC834 and for the evaluation of the raw data the program Calc415 (both Sunset Laboratory Inc., USA) were used. Further processing of the transmittance data was conducted with an external program (Microsoft Excel, Microsoft Corporation, USA).'*

- **In section 3.2, as least, the detection limit of $Fe_2O_3$ should be given by ICP-MS or OES.**

We agree that the limit of detection was missing for these methods and added them in line 126 in the revised manuscript. The limit of detection for ICP-MS and ICP-OES was 0.1 and 0.4 µgFe cm$^{-2}$, respectively.

- **Actually, section 3.4 and Figure S2 are nothing useful and can be deleted directly.**

The Figure (note that S2 changed to S3 in the revised version of the Supplement) presents the differences between the reference materials and the filter loaded with snow sample and explains why the correlations between iron loadings of the filters and $ATN_{700-400}$ (as presented in Figures 2 and 3 in the manuscript) are different. We included additional results of SEM and revised the text to explain the idea of the SEM analyses more clearly.

The new information is given in lines 256 - 262 and 269 - 271 of the revised manuscript:

'*We attribute the differences between the filters loaded with SRM 2709 and $Fe_2O_3$ shown in Fig. 2 to differences in the overall loadings of the filters (i.e. the loading with other material than hematite) and the particle sizes. Such differences are confirmed via SEM analyses of selected filters loaded with both reference materials. While $Fe_2O_3$ filters only contained hematite, SRM filers showed different kinds of minerals, i.e. particles consist of Ca, K, Si, Al, Mg, Na and O and just a minor contribution of Fe, which obviously leads to lower ATN values. Hematite particles on the $Fe_2O_3$ filters were generally larger in size compared to Fe-rich particulate matter in SRM 2709. Examples of the SEM analyses, elemental maps and spectra are given in the Supplement (Fig. S3).*'

'*The chemical composition of particulates in snow is more similar to SRM 2709 than to pure hematite, as shown in Fig. S3 in the Supplement. Still, SEM images show differences in particle size of SRM and snow filters, which might induce the differences together with differences in the chemical composition, which cannot be identified via SEM.*'

- **The caption of Figure S2 is unclear. Does Figure S2 is standard samples or observed samples?**

'$Fe_2O_3$ samples' refer to filters loaded with reference material throughout the manuscript. As more information was added to this Figure (Fig. S2 became Fig. S3 in the revised version), the caption was rewritten.

- **Finally, as Wang et al. (2012) indicated that the mineral dust mainly induces an extra decrease in optical reflectance during the 250 oC heating stage, thereafter, lead potential bias in the EC and OC split. But I didn't find any related illustration or explanation of such an issue in this study. I suggested the author should provide further details on this major issue of the split of EC and OC in snow samples to prove this useful approach.**

Figure 1(b) shows an example thermogram of a filter loaded with a snow sample that contained MD. The transmittance signal of the reruns, which cannot be attributed to carbonaceous compounds, shows a reduction at the 250°C, but even more pronounced reductions at higher temperatures. The aim of this manuscript is to highlight the influence of Fe on the transmittance logged during TOA and to deduce a method to quantify Fe loadings. Further recommendations for a correction of the split point are beyond the scope of this manuscript. This is actually a topic we are currently working on. To illustrate the topic of the OC/EC split in more detail, we included a thermogram containing the FID-signal in the Supplement and added a reference to this new Figure S1 in the main text (line 171 - 172 in the revised manuscript):

[Figure]

**Figure S1: Thermogram of a sample containing MD.**

*'The FID signal (green) shows higher amounts of OC in the sample than EC. The high FID signal caused by the calibration gas at the end of the measurement is not shown. An automatic split point could not be set for the sample, as the transmittance (scaled to be between 1390 and 8000 a.u.) does not reach its initial value during the measurement (black line). Using the transmittance at 250°C as a reference value for the split point would not yield EC concentrations, while a split point could be set using the transmittance at 550°C.'*

- **Finally, the author should note that there is potential mass loss of BC or MD on 1.0 µm quartz fiber filters compared with 0.4 µm Nuclepore filters, as shown in Figure 5 by Wang et al. (2020).**

The possibility of undercatch during filtration was mentioned in line 40 – 41; however, we thank the referee for the suggestion of the current work and added the citation in line 47 in the revised manuscript as well as in the References section in line 486 - 488. For TOA, no suitable alternative filter material to quartz fiber is available, as high temperature resistance and stable optical properties over a wide temperature range are indispensable. Thus, TOA with Nuclepore filters is not possible.

**References:**

**Wang M, Xu B, Zhao H, Cao J, Joswiak D, Wu G, et al. The Influence of Dust on Quantitative Measurements of Black Carbon in Ice and Snow when Using a Thermal Optical Method. Aerosol Science and Technology 2012; 46: 60-69.**

**Wang X, Zhang XY, Di WJ. Development of an improved two-sphere integration technique for quantifying black carbon concentrations in the atmosphere and seasonal snow. Atmospheric Measurement Techniques 2020; 13: 39-52.**

References:

Formenti, P., Caquineau, S., Desboeufs, K., Klaver, A., Chevaillier, S., Journet, E., & Rajot, J. L. (2014). Mapping the physico-chemical properties of mineral dust in western Africa: mineralogical composition. Atmospheric Chemistry and Physics, 14(19), 10663-10686.

Liu, H., Chen, T., Zou, X., Qing, C., & Frost, R. L. (2013). Thermal treatment of natural goethite: Thermal transformation and physical properties. Thermochimica Acta, 568, 115-121.

Wang, X., Zhang, X., & Di, W. (2020). Development of an improved two-sphere integration technique for quantifying black carbon concentrations in the atmosphere and seasonal snow. Atmospheric Measurement Techniques, 13(1), 39-52.